# Mind-Omni: A Unified Multi-Task Framework for Brain-Vision-Language Modeling via Discrete Diffusion

Yizhuo Lu [* 1 2]   Changde Du [* 1 3 4]   Qingyu Shi [5]   Hang Chen [1 2]   Jie Peng [1 3]   Liuyun Jiang [2]
Shuangchen Zhao [1 3 4]   Huiguang He [† 1 2 3 4]

## Abstract

Modeling the interplay between external stimuli and internal neural representations is a pivotal research area for Brain-Computer Interfaces (BCIs). A major limitation of prior work is the prevailing paradigm of specialized, single-task models, which curtails versatility and neglects inter-task synergies. To address this, we propose Mind-Omni, the first versatile framework that unifies seven distinct encoding and decoding tasks through a discrete diffusion paradigm. At its core is a novel Brain Tokenizer that transforms heterogeneous, continuous brain signals into standardized, discrete tokens. This enables direct, token-level interactions for mutual understanding and generation between any two or more modalities within a shared semantic space. To unlock advanced reasoning capabilities, we further curate a specialized Brain Question Answering (BQA) instruction-tuning dataset. Our model not only establishes a new state-of-the-art among multi-task unified frameworks but also provides strong evidence for multi-task synergy. By demonstrating performance competitive with, and at times superior to, larger specialized models, our work offers a powerful new paradigm for neural modeling and paves the way for foundation models of neural activity. The code is publicly available at https://github.com/ReedOnePeck/Mind-Omni.

[*]Equal contribution [1]NeuBCI Lab, State Key Laboratory of Brain Cognition and Brain-inspired Intelligence Technology, Institute of Automation, Chinese Academy of Sciences, Beijing, China [2]School of Future Technology, University of Chinese Academy of Sciences, Beijing, China [3]School of Artificial Intelligence, University of Chinese Academy of Sciences, Beijing, China [4]Zhongguancun Academy, Beijing, China [5]Peking University, Beijing, China. Correspondence to: Huiguang He <huiguang.he@ia.ac.cn>.

*Proceedings of the 43rd International Conference on Machine Learning*, Seoul, South Korea. PMLR 306, 2026. Copyright 2026 by the author(s).

## 1. Introduction

Recent breakthroughs in directly decoding visual images and linguistic content from brain activity [1,2,3,4,5,6,7,8,9, 10,11,12,13] have marked a milestone in Brain-Computer Interfaces and neuroscience [14,15,3,16,17]. However, these remarkable achievements predominantly rely on "expert models" highly optimized for a singular task (e.g., image reconstruction) or a single direction (e.g., decoding only or encoding only). This prevailing paradigm of specialization, illustrated in Tab. 1, inherently curtails model versatility and overlooks the profound synergistic potential among neural tasks. Consequently, developing a unified, multi-task neural encoding-decoding model is not only an urgent necessity to overcome current limitations but also a pivotal step towards building a "foundation model for the brain". This unified approach provides a powerful computational testbed for revealing the intrinsic relationships and mechanisms governing vision, language, and neural signals within a single framework.

*Table 1.* A comparison of the task capabilities of competing methods. Our unified model addresses seven tasks across different modalities including Image (I), Text (T), and Brain (B) signals (e.g., Brain-based Question Answering, BQA), in contrast to previous leading methods that are often specialized.

| Method | Subject | Encoding | | | Decoding | | | |
|---|---|---|---|---|---|---|---|---|
| | | I→B | T→B | I&T→B | B→I | B→T | B→I&T | BQA |
| MindEye[NeurIPS2024] [18] | Single | | | | ✓ | | | |
| MindBridge[CVPR2024] [19] | Multi | | | | ✓ | | | |
| Psychometry[CVPR2024] [20] | Multi | | | | ✓ | | | |
| BrainCap[NeurIPS2023 W] [21] | Single | | | | ✓ | ✓ | | |
| UMBRAE[ECCV2024] [22] | Multi | | | | | ✓ | | ✓ |
| Sem-Recons[Nat. Neurosci.] [23] | Single | | | | | ✓ | | |
| CLIP-Map[Nat. Mach. Intell.] [24] | Single | ✓ | | | | | | |
| MindSimulator[ICLR2025] [25] | Single | ✓ | | | | | | |
| BraVL[TPAMI2023] [26] | Single | ✓ | ✓ | ✓ | ✓ | | | |
| Ours | Multi | ✓ | ✓ | ✓ | ✓ | ✓ | ✓ | ✓ |

However, constructing such a unified model presents substantial challenges. (1) **Input Heterogeneity**: Inter-subject anatomical variability prevents direct integration of brain signals, while existing alignment methods either scale poorly with subject count [27,28] or sacrifice information fidelity [19]. (2) **Modality Disparity**: The semantic gap between continuous brain signals and discrete visual/textual tokens remains a persistent barrier, making direct cross-modal fusion ineffective [29,30,31]. (3) **Task Interdependence**: The relationships between different encoding and decod-

ing tasks are poorly understood. While specialized models assume task independence, we hypothesize that shared neural representations could enable positive transfer between related tasks—such as simultaneous image and text decoding—but this requires a unified architecture to test.

To address these challenges, we introduce Mind-Omni, the first framework to concurrently handle seven distinct encoding and decoding tasks. Our approach introduces three key solutions: (1) **Standardized Neural Representation**: We adopt MNI152 standard space registration [32,33,34,35] to establish consistent structural-space dimensionality across subjects, providing a scalable foundation for subsequent processing and mitigating the need for subject-specific modules. (2) **Semantically-Aligned Brain Tokenization**: We introduce a novel Brain Tokenizer that transforms continuous fMRI signals into discrete tokens aligned with image and text representations in a shared semantic space. This is achieved through a multi-level alignment strategy that balances reconstruction fidelity with cross-modal semantic consistency. (3) **Unified Discrete Diffusion Framework**: Rather than composing separate task-specific modules, we leverage discrete diffusion modeling to create a single generative process that unifies all seven tasks. This architecture choice is crucial—unlike autoregressive models [36,37,38] that impose sequential dependencies, the permutation-invariant nature of diffusion models [39,40,41,42] is vital for observing cross-task synergies without the confounding bias of a prescribed generation order. Furthermore, to unlock the framework's advanced reasoning capabilities, we curate a novel Brain Question Answering instruction-tuning dataset by leveraging Qwen2-VL (7B) model [43] and LLaVA-Instruct-150K [44,45].

**Our investigations further reveal novel insights into multi-task learning for neural signals.** We found that (1) there are complementary effects between vision and language modalities within the encoding task; (2) a synergistic relationship exists between different decoding objectives (e.g., B→T and B→I), where concurrent training and inference enhance performance on both.

In summary, the contributions of this paper are threefold:

- **Paradigm Shift in Neural Modeling:** We propose Mind-Omni, the first framework to unify seven encoding and decoding tasks across brain, image, and text within a single discrete diffusion architecture, moving beyond the specialized model paradigm that has dominated the field.
- **Novel Cross-Modal Fusion Mechanism:** We introduce a Brain Tokenizer that bridges the modality gap between neural signals and visual-textual data, enabling direct, token-level interaction in a shared semantic space.
- **Demonstrated Multi-Task Synergy:** We provide evidence for synergistic effects in joint neural modeling.

Mind-Omni not only establishes a new SOTA for unified frameworks but also competes with, and sometimes surpasses, larger specialized models, validating the efficacy of our unified approach.

**Conflict of Interest Disclosure.** The authors declare that they have no financial conflicts of interest related to this work.

## 2. Related Work

### 2.1. Neural Encoding and Decoding

Recent years have seen a surge of impressive research in neural encoding [46,47,48,49] and decoding [50,51,52,13, 53,54]. Many state-of-the-art methods, from the linear encoding models of Wang et al. [24] and the diffusion-based mapping of Bao et al. [25] to the image reconstruction techniques of Takagi [9], Ozcelik et al. [55], and Scotti et al. [18,27], adopt a "feature mapper + pre-trained model" paradigm (Fig. 1(a, b)). This paradigm also extends to language, where Xia [22], Shen et al. [56] used a feature projector to connect brain signals with LLMs for question-answering. However, a fundamental limitation of this two-stage approach is its reliance on separate, frozen generative models. This structure not only constrains models to singular tasks but, more critically, prohibits deep, reciprocal fusion between neural activity and external stimuli.

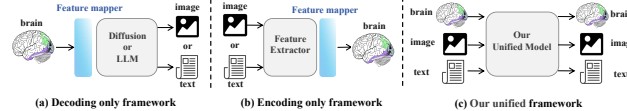

*Figure 1.* Architectural comparison of our framework against specialized models for neural encoding or decoding. We replace the standard feature mapper plus pre-trained model pipeline with a direct multi-modal fusion of image, text, and brain activity.

In contrast, our work presents a unified framework that obviates the need for external specialist models. Our approach aligns with the joint modeling paradigm of MoPoE [57] and BraVL [26] (Fig. 1(c)), but advances it by using discrete diffusion paradigm, enabling the integration of complex language tasks like question-answering within a single, cohesive framework.

### 2.2. Unified MLLMs For Generation and Understanding

A prominent line of research in MLLMs has focused on unifying image-text understanding and generation [58,59]. These efforts span several paradigms: (1) autoregressive models, such as the Emu series [36,37,38], Seed-X [60], and Chameleon [61]; (2) discrete diffusion models, including Mmada [62] and Muddit [39]; and (3) hybrid architectures like Show-O [63] and Transfusion [64]. While powerful, the fixed causal structure of autoregressive models conflicts with the concurrent nature of multi-modal encoding/decoding, thereby precluding an unbiased study of synergistic rela-

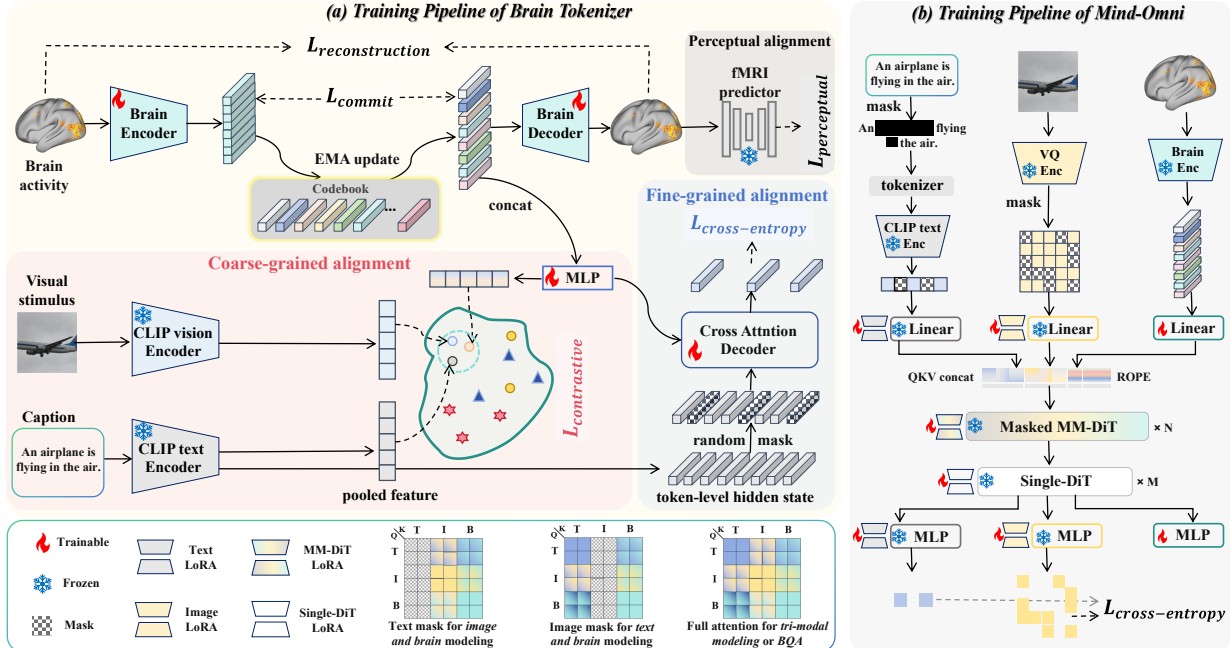

*Figure 2.* Training pipeline for our proposed framework, featuring a Brain Tokenizer and a DiT-based discrete diffusion model. (a) The Brain Tokenizer discretizes continuous fMRI signals into a sequence of tokens. It is trained with a composite objective that includes reconstruction and commitment losses, supplemented by coarse- and fine-grained modality alignment losses, as well as a perceptual alignment loss. (b) The diffusion model is then trained on a masked prediction objective, illustrated here with the B→I&T task. It learns to restore masked image and text tokens during the denoising process using a cross-entropy loss. We also introduce specific modality masking schemes (bottom left) in the masked MM-DiT to handle cases of missing modalities during training and inference (See Fig. 3).

tionships. We therefore align with the discrete diffusion paradigm for their permutation-invariant nature. We thus initialize our model with the parameters of Muddit [39], building upon its proven foundation for unified modeling. See Appendix A for a detailed discussion.

## 3. Method

Our framework is comprised of two main components. First, we employ tokenizers to discretize the continuous signals from three distinct modalities: image, text, and brain signals. Specifically, we utilize a pre-trained VQ-VAE [65] for the image modality and the CLIP tokenizer [66] for the text modality. For brain signals, we trained a Brain Tokenizer that aligns with the shared semantic space (Section 3.1). Second, building upon the bi-modal backbone pre-trained in Muddit [39], we extend the architecture by incorporating a third branch for brain responses. This extended framework leverages discrete diffusion modeling to unify the training and sampling processes across seven distinct neural encoding and decoding tasks (Sections 3.2 and 3.3).

### 3.1. Brain Tokenizer

To discretize continuous functional magnetic resonance imaging (fMRI) signals into meaningful tokens for the diffusion model, we introduce a specialized Brain Tokenizer. As depicted in Fig. 2(a), its architecture comprises a VQ-VAE-

style backbone augmented with three alignment strategies to bridge the significant modality gap between brain signals and visual-textual representations.

Let the training data consist of paired samples $\{(\mathbf{b}_i, \mathbf{v}_i, \mathbf{c}_i)\}_{i=1}^{B}$, where $\mathbf{b}_i \in \mathbb{R}^{1 \times N_{voxel}}$ is an fMRI recording, $\mathbf{v}_i$ is the corresponding visual stimulus, and $\mathbf{c}_i$ is its textual caption. The backbone of the Brain Tokenizer consists of an encoder $E_{brain}$ (composed of 1D CNN and MLP layers), a decoder $D_{brain}$, and a discrete codebook $\mathcal{Z} \in \mathbb{R}^{K \times D}$, where $K$ is the codebook size. The fMRI signal $\mathbf{b}$ is first mapped to a sequence of discrete tokens $\mathbf{z}_b \in \mathbb{R}^{B \times D}$ and then reconstructed as $\hat{\mathbf{b}} \in \mathbb{R}^{B \times N_{voxel}}$. The training of this backbone is supervised by a reconstruction loss and a commitment loss. The codebook is updated via an Exponential Moving Average (EMA).

$$L_{VQ} = \|\mathbf{b} - \hat{\mathbf{b}}\|_2^2 + \beta \|\text{sg}[E_{brain}(\mathbf{b})] - \mathbf{z}_b\|_2^2, \quad (1)$$

where $\text{sg}[\cdot]$ denotes the stop-gradient operator.

To ensure the learned brain tokens are semantically aligned with visual and textual data, we introduce the following alignment objectives:

**Coarse-grained Alignment.** We first perform a coarse-grained alignment in the shared semantic space of CLIP-H. The discrete fMRI tokens $\mathbf{z}_b$ are concatenated and

passed through an MLP to produce a global fMRI feature $\mathbf{f}_b \in \mathbb{R}^{B \times 1024}$. We then employ a tri-modal contrastive loss to pull $\mathbf{f}_b$ closer to its corresponding image CLIP feature $\mathbf{f}_v \in \mathbb{R}^{B \times 1024}$ and text CLIP feature $\mathbf{f}_c \in \mathbb{R}^{B \times 1024}$ while pushing it away from non-corresponding pairs. To stabilize the training, we supplement this with a feature distillation loss, which directly minimizes the Mean Squared Error (MSE) between the brain feature $\mathbf{f}_b$ and the visual feature $\mathbf{f}_v$. The combined coarse-grained alignment loss is:

$$L_{\text{coarse-grain}} = L_{\text{InfoNCE}}(\mathbf{f}_b, \mathbf{f}_c) + L_{\text{InfoNCE}}(\mathbf{f}_b, \mathbf{f}_v) + \|\mathbf{f}_b - \mathbf{f}_v\|_2^2, \tag{2}$$

**Fine-grained Alignment.** For a more granular alignment at the token level, we leverage the token-level hidden states of the text caption, $\mathbf{h}_c \in \mathbb{R}^{B \times 77 \times 1024}$, extracted from the CLIP text encoder. We randomly mask a portion of (always 30%) these text tokens and task a cross-attention decoder to predict the original masked tokens. In this module, the masked text tokens serve as the Query, while the sequence of fMRI tokens $\mathbf{f}_b$ serves as the Key and Value. The alignment is optimized via a cross-entropy loss.

$$L_{\text{fine-grain}} = \text{CrossEntropy}(D_{\text{cross\_attn}}(\mathbf{h}_c^{\text{masked}}, \mathbf{f}_b), \mathbf{h}_c). \tag{3}$$

Therefore, the total semantic alignment loss is formulated as:

$$L_{\text{SA}} = \lambda_1 L_{\text{coarse-grain}} + \lambda_2 L_{\text{fine-grain}}. \tag{4}$$

**Perceptual Alignment.** To ensure the reconstructed fMRI signal is semantically meaningful, not just structurally similar, we posit that $\hat{\mathbf{b}}$ must be decodable into its corresponding CLIP features. To this end, we employ a pre-trained and frozen fMRI predictor $P_{\text{fMRI}}$, which is an MLP trained to map ground-truth fMRI signals to their corresponding image and text CLIP features. We enforce perceptual alignment by ensuring that the CLIP features predicted from our reconstructed fMRI $\hat{\mathbf{b}}$ are close to the ground-truth CLIP features $(\mathbf{f}_v, \mathbf{f}_c)$.

$$L_{\text{perceptual}} = \|P_{\text{fMRI}}(\hat{\mathbf{b}})_v - \mathbf{f}_v\|_2^2 + \|P_{\text{fMRI}}(\hat{\mathbf{b}})_c - \mathbf{f}_c\|_2^2, \tag{5}$$

where $P_{fMRI}(\cdot)_v$ and $P_{fMRI}(\cdot)_c$ denote the predicted image and text features, respectively.

Finally, the total loss for training the Brain Tokenizer is a weighted sum of the aforementioned components:

$$L = L_{\text{VQ}} + L_{\text{SA}} + \lambda L_{\text{perceptual}}, \tag{6}$$

where $\lambda, \lambda_1,$ and $\lambda_2$ are hyperparameters that balance the contributions of each alignment objective.

### 3.2. Unified Training of Mind-Omni

After obtaining discrete tokens for the image, text, and brain modalities via their respective pre-trained tokenizers, we adapt the original Muddit [39] to handle tri-modal inputs. Specifically, we first augment the overall architecture with a dedicated input layer and an output layer for the brain tokens, as shown in Fig. 2(b). Then, to enable deep fusion within the core generative block, we modify the MM-DiT by symmetrically adding dedicated Q, K, V projection and normalization layers for the new brain modality branch. This extension mirrors the architecture of the existing image and text branches, while the underlying Single-DiT blocks remain architecturally unchanged.

Our framework unifies seven distinct neural encoding and decoding tasks within a single training paradigm. The core principle is to frame every task as a conditional masked token prediction problem under the discrete diffusion framework.

**Unified Training Objective.** Let $\mathbf{x}^I, \mathbf{x}^T, \mathbf{x}^B$ represent the token sequences for the image, text, and brain modalities, respectively. During training, we apply the forward corruption process to the *target* modalities by stochastically replacing their tokens with a special [MASK] token. The proportion of masked tokens, known as the mask ratio $\gamma_t$, is determined by a time-dependent schedule.

Following recent advancements in generative modeling [67, 68, 69], we adopt a cosine scheduling strategy. A timestep $t \in [0, 1]$ is sampled from a truncated arccos distribution, which emphasizes sampling timesteps that correspond to intermediate mask ratios. The probability density function for sampling $t$ is given by:

$$p(t) = \frac{2}{\pi} \left(1 - (1-t)^2\right)^{-\frac{1}{2}}. \tag{7}$$

The corresponding survival probability $\alpha_t$—the probability that a token remains unmasked at time $t$—is then defined by the cosine schedule as $\alpha_t = \cos(0.5\pi t)$. The mask ratio is thus $\gamma_t = 1 - \alpha_t$. Our tri-modal backbone, denoted as $\mathtt{G}$, is subsequently trained to predict the original, uncorrupted tokens of the target modalities given the corrupted target tokens and the clean conditioning tokens.

This is optimized via a unified, continuous-time negative ELBO loss, which is shared across all seven tasks. Let $\mathcal{T}$ be the set of target modalities and $\mathcal{C}$ be the set of conditioning modalities for a given task. For brevity, we denote the set of corrupted tokens from all target modalities as $\mathbf{X}_t^{\mathcal{T}} = \{\mathbf{x}_t^{m'}\}_{m' \in \mathcal{T}}$ and the set of clean tokens from all conditioning modalities as $\mathbf{X}_0^{\mathcal{C}} = \{\mathbf{x}_0^{m''}\}_{m'' \in \mathcal{C}}$. The unified loss is then formulated as:

$$\mathcal{L}_{\text{unified}} = \sum_{m \in \mathcal{T}} \mathbb{E}_{q(\mathbf{x}_t^m | \mathbf{x}_0^m)} \left[ \int_0^1 \frac{\alpha_t'}{1 - \alpha_t} \log \left( \mathtt{G}(\mathbf{X}_t^{\mathcal{T}}, \mathbf{X}_0^{\mathcal{C}}, t)_m \cdot \mathbf{x}_0^m \right) dt \right] \tag{8}$$

where $\mathbf{x}_0^m$ are the ground-truth tokens for a target modality $m \in \mathcal{T}$, $\mathbf{x}_t^m$ are its corrupted version at time $t$, and $\mathtt{G}(\cdot)_m$

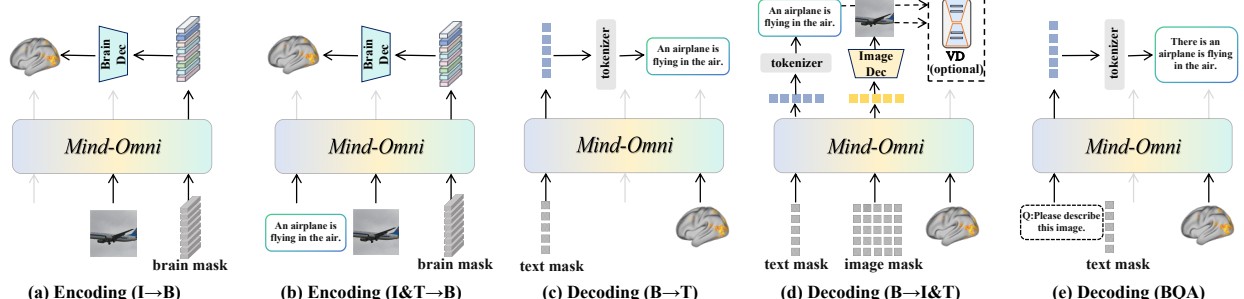

*Figure 3.* Model inference pipeline, where masked tokens serve as the input for the target modality. (a,c) For single-modality encoding or decoding tasks, specific modality masking schemes are employed to maintain consistent input dimensionality. (e) For the BQA task, the text-side input is a concatenation of question tokens and a sequence of mask tokens that serve as a placeholder for the answer.

denotes the model's prediction specifically for modality $m$. **The key insight is that by strategically defining the target sets $\mathcal{T}$ and conditioning sets $\mathcal{C}$, this single objective function can steer the training for all desired tasks.** Unused modalities in any given task are handled via attention masking within the MM-DiT block.

**Task-Specific Masking Strategies.** Each of our seven tasks is realized by specifying its conditioning ($\mathcal{C}$) and target ($\mathcal{T}$) sets. For instance, in joint decoding (B $\rightarrow$ I&T, Fig. 2(b)), the brain modality is the condition ($\mathcal{C} = \{B\}$), while image and text are treated as joint targets ($\mathcal{T} = \{I, T\}$), whose tokens are corrupted synchronously using a shared timestep and mask ratio. **This synchronous masking strategy is crucial for enabling and observing the synergistic effects in joint image-text decoding.** For single-modality encoding (T $\rightarrow$ B), text is the condition ($\mathcal{C} = \{T\}$) and brain is the target ($\mathcal{T} = \{B\}$), where a specific attention mask isolates the unused image modality. The framework also supports complex tasks like Brain Question Answering (BQA), where the condition combines brain and question tokens ($\mathcal{C} = \{B, T_{question}\}$) to generate the answer ($\mathcal{T} = \{T_{answer}\}$). The remaining tasks are defined analogously, allowing our single generator G to unify all seven objectives.

### 3.3. Unified Inference of Mind-Omni

Inference across all seven tasks is performed via a unified, iterative denoising procedure that reverses the forward corruption. Starting with fully masked token sequences for all target modalities, we progressively recover the original tokens over $T$ discrete timesteps. The core of the inference is the sampling step $\mathbf{x}_{t-\frac{1}{T}} = \mathtt{S}(\mathbf{x}_t, t)$, applied from $t = 1, \frac{T-1}{T}$ down to $\frac{1}{T}$. At each step, tokens that have already been revealed remain unchanged. Conversely, any token $x_t$ that is a [MASK] is resampled from a categorical distribution. The probability vector for this new token, $p(\mathbf{x}_{t-\frac{1}{T}})$, is a convex combination of the [MASK] token $\mathbf{m}$ and the model's prediction of the clean token, $\hat{\mathbf{x}}_0 = \mathtt{G}(\mathbf{X}_t^{\mathcal{T}}, \mathbf{X}_0^{\mathcal{C}}, t)$, given

by:

$$p(\mathbf{x}_{t-\frac{1}{T}}) = \frac{(1 - \alpha_{t-\frac{1}{T}})\mathbf{m} + (\alpha_{t-\frac{1}{T}} - \alpha_t)\hat{\mathbf{x}}_0}{1 - \alpha_t}. \quad (9)$$

As illustrated in Fig. 3, the specific task is determined simply by how the initial token sequences are prepared. For example, to perform joint image-text decoding (B $\rightarrow$ I&T), we initialize the image and text modalities as fully masked sequences ($\mathbf{x}_1^I, \mathbf{x}_1^T$), while providing the ground-truth brain tokens as the condition ($\mathbf{x}_0^B$). The sampler then iteratively applies the reverse update rule for $T$ steps to jointly generate the final clean tokens, $\mathbf{x}_0^I$ and $\mathbf{x}_0^T$, which are then passed to their respective decoders. All other encoding and decoding tasks follow this exact same procedure, merely by reassigning which modalities serve as the condition versus the targets to be initialized with masks. For the foundations of discrete diffusion models, see Appendix B.

## 4. Experimental Setup

**Data and Preprocessing.** We train our model on fMRI data from 8 subjects (40 sessions) of the Natural Scenes Dataset (NSD) [70]. For evaluation, we test on subjects 1, 2, 5, and 7, consistent with prior work [18]. To address inter-subject variability and ensure consistent input dimensions, we register all functional data to the MNI152 standard template [33,34,35] using FSL [32]. We then extract and flatten the voxels corresponding to the visual cortex. Additionally, we curate a Brain Question Answering (BQA) dataset for instruction tuning, leveraging Qwen2-VL [43] and LLaVA-Instruct-150K [44,45], which includes short and long-form descriptions, and reasoning tasks (details in Appendix C).

**Progressive Training.** To stabilize training on such diverse tasks, we employ a progressive protocol that first establishes robust brain-modality alignment before jointly fine-tuning the entire model for high-level reasoning. In Stage 1, we first freeze the pre-trained Muddit [39] backbone and train only our newly introduced parameters on joint encoding (I&T $\rightarrow$ B) and decoding (B $\rightarrow$ I&T) tasks. Subsequently, we perform multi-task training across all six

single- and bi-modal objectives to establish comprehensive inter-modal translation capabilities. Finally, in Stage 2, we fine-tune the backbone using DoRA [71] and introduce the BQA dataset to unlock its reasoning capabilities. (the training configurations are detailed in Appendix D).

# 5. Results

## 5.1. Holistic Quantitative Assessment

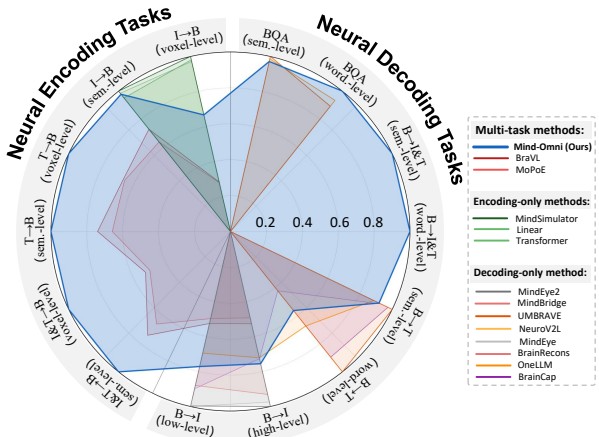

*Figure 4.* Holistic comparison against specialized SOTAs. Metrics are aggregated and max-normalized (1.0 = SOTA performance).

We begin with a comprehensive evaluation of our model against previous specialized SOTAs (e.g., MindEye2 [27], MindSimulator [25]) across seven neural encoding and decoding tasks. As shown in Fig.4, task-specific metrics are aggregated via averaging and normalized by their maximum values. While our model lags behind the corresponding single-task specialists on a few tasks such as B→I and B→T, the comparison highlights its unique versatility: unlike specialists limited to single domains, our unified model maintains competitive performance across the full spectrum. This capability is further corroborated by the qualitative results in Fig.5. **For a direct and fair comparison under a consistent multi-task setting, the following analyses will focus on unified models of similar nature, such as BraVL [26] and MoPoE [57].**

**Neural Decoding.** Table 2 presents the quantitative results for visual reconstruction. With a more compact parameterization, our model achieves competitive performance across more tasks, establishing a new SOTA among unified models. Additional visual reconstruction results are provided in Appendix E. In Table 3, we compare our model directly against the specialized language-decoding models UMBRAE [22] and OneLLM [72]. The results demonstrate our model's dominant SOTA performance on the challenging Detail Description and Reasoning tasks. Notably, while both UMBRAE and OneLLM depend on external LLMs such as Vicuna-13B [73], our framework is self-contained, built upon Muddit-1B [39], and does not rely on such ex-

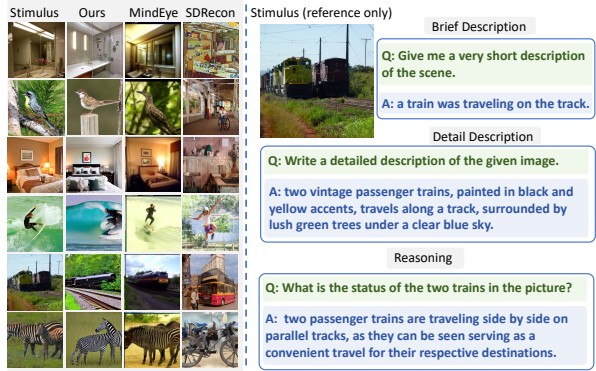

*Figure 5.* Results of neural decoding. Left: Image reconstruction. Right: Brain question answering. See Appendix E for more results.

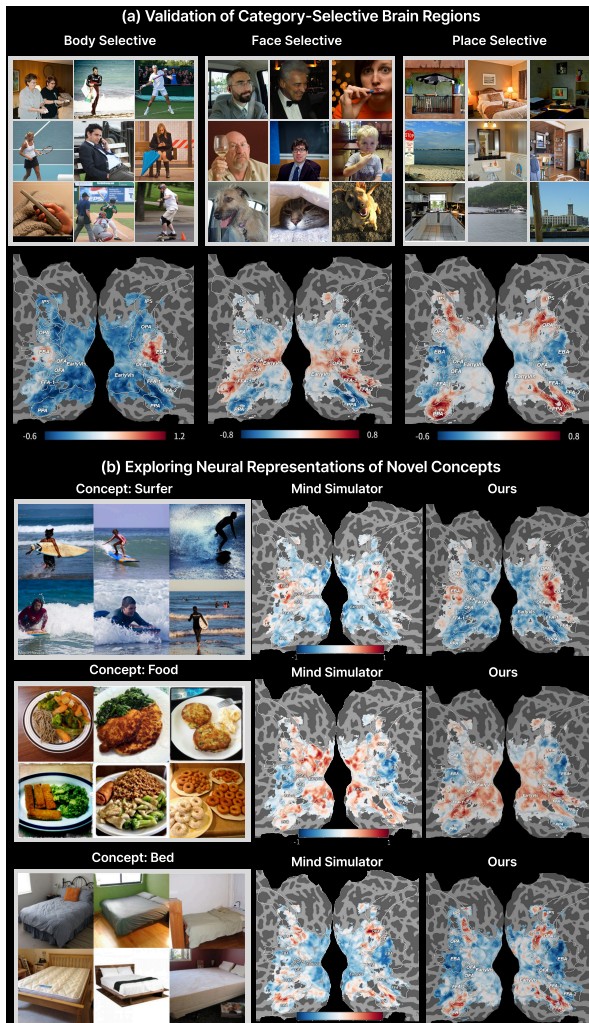

*Figure 6.* Mind-Omni as a computational testbed. (a) Replicating established category-selective regions. (b) Probing cortical topography for novel concepts. See Appendix I for more results.

*Table 2.* Quantitative evaluation of visual reconstruction (B→I and B→I&T tasks) averaged over subjects 1, 2, 5, and 7. The best, and second-best results among comparable models are highlighted in **bold**, and with an underline, respectively.

| Method | Trainable Parameters | # Models | # Tasks | Low-Level | | | | High-Level | | | |
|---|---|---|---|---|---|---|---|---|---|---|---|
| | | | | PixCorr ↑ | SSIM ↑ | AlexNet(2) ↑ | AlexNet(5) ↑ | Inception ↑ | CLIP ↑ | EffNet-B ↓ | SwAV ↓ |
| SDRecon [9] | 3B | 4 | 2 | – | – | **83.0%** | 83.0% | 76.0% | 77.0% | – | – |
| MoPoE [57] | 564M | 4 | 4 | .021 | .145 | 54.4% | 56.2% | 59.3% | 57.6% | .965 | .752 |
| BraVL [26] | 564M | 4 | 4 | .023 | .167 | 56.3% | 59.1% | 61.7% | 63.5% | .943 | .757 |
| OneLLM [72] | 7B | 4 | 7 | .053 | .313 | 64.7% | 76.1% | 75.3% | 77.2% | .851 | .551 |
| Ours (B→I) | 442M | 1 | 7 | **.118** | **.383** | 67.1% | 72.8% | 69.4% | 66.7% | .918 | .583 |
| Ours (B→T) | 442M | 1 | 7 | .036 | .284 | 62.8% | 70.4% | 67.9% | 69.9% | .895 | .623 |
| Ours (B→I&T) | 442M | 1 | 7 | .058 | .341 | 72.5% | **84.9%** | **78.8%** | **79.8%** | .824 | .537 |

*Table 3.* Quantitative analysis of detailed descriptions (B→T and B→I&T tasks), and reasoning (BQA task). The best and second-best results are indicated in **bold** and with an underline, respectively. LLM-as-Judge refers to evaluation using Qwen3-VL-30B-A3B-FP8: given the stimulus image, question, reference answer, and model output, the judge determines whether the answer is correct.

| Method | Train. Params | External LLM | # Models | # Tasks | BLEU1 ↑ | BLEU2 ↑ | BLEU3 ↑ | METEOR ↑ | ROUGE ↑ | CIDEr ↑ | SPICE ↑ | CLIP ↑ | RefCLIP ↑ | BERT ↑ | LLM-as-Judge ↑ |
|---|---|---|---|---|---|---|---|---|---|---|---|---|---|---|---|
| | | | | | **Detail Description** | | | | | | | | | | |
| UMBRAE [22] | 146.57M | Vicuna (13B) [73] | 1 | 2 | 21.39 | 11.86 | 6.31 | 11.31 | 17.60 | 6.04 | 6.43 | **60.85** | **65.96** | – | – |
| OneLLM [72] | 7B | LLaMA2 (7B) [74] | 4 | 7 | 18.41 | 12.13 | 5.34 | 9.37 | 17.13 | 9.41 | 6.34 | 50.31 | 51.43 | **86.18** | – |
| Ours (B→T) | 442M | None | 1 | 7 | 14.92 | 9.03 | 5.83 | 13.86 | 13.35 | 6.28 | 6.78 | 48.47 | 47.00 | 80.21 | – |
| Ours (B→I&T) | 442M | None | 1 | 7 | **29.12** | **17.63** | **11.36** | **26.05** | **30.54** | **12.26** | **13.25** | 53.67 | 52.75 | **87.73** | – |
| | | | | | **Reasoning** | | | | | | | | | | |
| UMBRAE [22] | 146.57M | Vicuna (13B) [73] | 1 | 2 | **46.33** | **31.42** | **23.56** | 41.93 | 42.12 | 156.81 | 33.70 | 69.57 | 75.22 | 91.33 | **25.48** |
| OneLLM [72] | 7B | LLaMA2 (7B) [74] | 4 | 7 | 19.27 | 13.77 | 10.76 | 42.83 | 45.63 | 223.67 | 37.43 | 67.46 | 73.41 | **91.93** | 19.12 |
| Ours (BQA) | 442M | None | 1 | 7 | 23.18 | 15.83 | 11.86 | **50.13** | **52.91** | **223.98** | **43.28** | **70.65** | **76.72** | 81.96 | 24.37 |

*Table 4.* Evaluation of neural encoding via PCC, MSE, and RSA in voxel and semantic spaces. **Bold** and underline indicate the best and second-best performance.

| Method | Voxel-Level | | | Semantic-Level | | |
|---|---|---|---|---|---|---|
| | gPCC↑ | gMSE↓ | gRSA↑ | $PCC_{semantic}$↑ | $MSE_{semantic}$↓ | $RSA_{semantic}$↑ |
| MoPoE [57] | 0.085 | 0.845 | 0.215 | 0.468 | 0.386 | 0.382 |
| BraVL [26] | 0.083 | 0.823 | 0.221 | 0.523 | 0.299 | 0.417 |
| Ours (T→B) | 0.108 | 0.729 | 0.307 | 0.738 | 0.198 | 0.619 |
| Ours (I→B) | 0.157 | 0.656 | 0.403 | 0.679 | 0.224 | 0.562 |
| Ours (I&T→B) | **0.160** | **0.654** | **0.408** | **0.754** | **0.187** | **0.654** |

ternal components. Its superior performance on these tasks suggests a deeper understanding grounded in the brain signals themselves, rather than in external knowledge priors. More language decoding results are provided in Fig. 13.

**Neural Encoding.** Table 4 quantitatively assesses Mind-Omni's performance on single-modal (I→B, T→B) and multi-modal (I&T→B) neural encoding tasks. We evaluate predictions in both the high-dimensional voxel space (∼13,000 dimensions) and a projected CLIP semantic space (1024 dimensions), employing Pearson Correlation Coefficient (PCC) and Mean Squared Error (MSE) for local similarity, and Representational Similarity Analysis (RSA) [75] for global structural alignment. Our model consistently outperforms the multi-task baselines BraVL [26] and MoPoE [57] across all metrics. As indicated in Fig. 4, while our model trails the specialized single-task model MindSimulator [25] on voxel-level metrics, it achieves comparable performance at the semantic level. This suggests that although the vector quantization step incurs some loss of fine-grained voxel-wise accuracy, it enables the model to capture richer semantic representations in visual cortex—a capability that is further validated by the semantic selectivity analysis in the next section (Fig. 6).

## 5.2. Mind-Omni as a Computational Testbed

We adopt conceptual representation as a case study to demonstrate Mind-Omni's potential as a computational tool for neuroscientific exploration. As shown in Fig. 6(a), we first validate the model by successfully replicating well-established category-selectivity in visual cortex for bodies, faces, and places. Using images from the NSD test split (approx. 50 per category), we generate predicted fMRI responses and project them onto the cortical surface. The results exhibit localized and subject-consistent activations in the expected functional regions (EBA for bodies, OFA/FFA for faces, PPA/OPA for scenes). Results across additional subjects are provided in Appendix I. This confirms that Mind-Omni captures high-level functional architecture beyond superficial numerical fitting.

Building on this validation, we further employ the model to probe novel concept-selective regions (e.g., "Surfer"). Following the methodology of MindSimulator [25], we select concept-specific images from MSCOCO and synthesize their corresponding fMRI responses. The resulting cortical maps align closely with those from MindSimulator, demonstrating comparable effectiveness in capturing concept-level information. Notably, this observation aligns with the principle of distributed neural processing, where complex concepts are represented not by isolated voxels but by the coordinated activity of spatially distributed brain regions [76,77,78]. Together, these results underscore Mind-Omni's utility as a tool for investigating the organization and distribution of semantic representations in the brain.

# 6. Towards a Better Unified Model

As the first framework to unify seven distinct neural encoding and decoding tasks, our work raises a critical question: how can such a model be constructed effectively? In this section, we distill critical design principles and conduct a systematic investigation into three core aspects: (1) the architectural design of the core modality-bridging component, (2) the optimization strategy required for stable multi-task learning, and (3) the emergent synergistic properties that validate the unified approach itself.

## 6.1. The Architecture Design of Brain Tokenizer

*Table 5.* Ablation study on the Brain Tokenizer's architecture design, where rPCC is calculated on self-reconstructed fMRI signals. The chance level for retrieval is 0.05.

| $\mathcal{L}_{SA}$ | $\mathcal{L}_{perceptual}$ | codebook size | code dim. | code num. | rPCC ↑ | Retrieval (Top50) ↑ | | codebook usage ↑ |
|---|---|---|---|---|---|---|---|---|
| | | | | | | B2I | B2T | |
| × | × | 64 | 512 | 64 | 0.37 | 0.05 | 0.05 | 1% |
| × | × | 64 | 128 | 64 | 0.39 | 0.05 | 0.05 | 6% |
| × | × | 64 | 16 | 64 | 0.43 | 0.05 | 0.05 | 70% |
| × | × | 128 | 16 | 64 | 0.45 | 0.05 | 0.05 | 32% |
| ✓ | × | 64 | 16 | 64 | 0.64 | 0.58 (+0.53) | 0.54 (+0.49) | 100% (+30%) |
| ✓ | × | 128 | 16 | 64 | 0.68 | 0.60 | 0.57 | 62% |
| ✓ | × | 128 | 32 | 32 | 0.64 | 0.61 | 0.58 | 38% |
| ✓ | × | 256 | 16 | 64 | 0.63 | 0.60 | 0.57 | 40% |
| ✓ | × | 384 | 16 | 64 | 0.64 | 0.60 | 0.56 | 35% |
| ✓ | × | 512 | 16 | 64 | 0.62 | 0.58 | 0.53 | 28% |
| ✓ | ✓ | 128 | 16 | 64 | **0.68** | 0.62 (+0.02) | 0.59 (+0.02) | 80% (+18%) |
| ✓ | ✓ | 128 | 32 | 32 | 0.64 | **0.68** (+0.07) | **0.64** (+0.06) | 40% (+2%) |

We first dissect the architectural design and loss components of our Brain Tokenizer (Tab. 5), yielding several key insights. (1) Unlike tokenizers for natural images, the lower intrinsic dimensionality of fMRI signals makes the model prone to codebook collapse when the codebook is over-parameterized in size or dimension. (2) The semantic alignment loss ($\mathcal{L}_{SA}$) proves critical, exhibiting a dual benefit: it substantially boosts retrieval performance from a chance-level of 0.05 to 0.58—vital for downstream task alignment—while also markedly improving self-reconstruction fidelity (rPCC from 0.43 to 0.64) and codebook usage (+30%). (3) The perceptual alignment loss ($\mathcal{L}_{perceptual}$) further enhances codebook usage, thereby increasing its richness and diversity. Ablations on the specific contributions of the coarse- and fine-grained alignment losses are deferred to the Appendix G.

## 6.2. The Role of Training Strategy in Unification

We then ablate key aspects of our training strategies (Tab. 6). First, we find that a tokenization strategy that increases the number of tokens while reducing their individual dimension enhances encoding performance without compromising decoding quality (a). The results in (b) underscore the necessity of our progressive training curriculum, as a naive joint training approach leads to marked performance degradation. Furthermore, training from scratch without the pre-trained Muddit priors proves similarly detrimental, highlighting the criticality of leveraging existing knowledge in the data-constrained fMRI regime. Finally, (c) confirms

that data enrichment through fine-grained captions curated with Qwen2-VL yields consistent gains.

*Table 6.* Ablation study on the impact of different training strategies. Blue-shaded row denote the strategy used in our model. Complete results are provided in Tabs. 13−18.

| Task | Decoding | | | Encoding | | |
|---|---|---|---|---|---|---|
| | BLEU1↑ | ROUGE↑ | BERT↑ | gPCC↑ | gMSE↓ | gRSA↑ |
| **(a) Choice of Tokenization Strategy** | | | | | | |
| code dim.=32 | 29.56 | **25.85** | **88.32** | 0.126 | **0.621** | 0.315 |
| code dim.=16 | **30.28** | 25.73 | 88.04 | **0.145** | 0.698 | **0.342** |
| **(b) Choice of Training Strategy** | | | | | | |
| Direct | 29.11 | 27.31 | 84.29 | 0.132 | 0.677 | 0.369 |
| Progressive | **29.12** | **30.54** | **87.73** | **0.160** | **0.654** | **0.408** |
| From Scratch | 20.35 | 14.32 | 74.66 | 0.104 | 0.984 | 0.242 |
| From Pretrained | **29.12** | **30.54** | **87.73** | **0.160** | **0.654** | **0.408** |
| **(c) Choice of Image-Caption Pairs** | | | | | | |
| Raw COCO | 24.37 | 26.52 | 83.17 | 0.133 | 0.671 | 0.384 |
| Qwen2-VL Enhanced | **29.12** | **30.54** | **87.73** | **0.160** | **0.654** | **0.408** |

## 6.3. Synergistic Gains in the Multi-task Framework

**Inter-modal Complementarity.** For neural encoding, we observe a distinct complementary effect between modalities. As shown in Tab. 4, the joint-modality condition (I&T → B) consistently outperforms single-modality inputs across all metrics. Fig. 7 further illustrates that while image-only encoding activates both early and high-level visual areas, and text-only encoding primarily engages semantic regions, the joint model achieves high accuracy across the entire visual cortex—suggesting it captures the brain's natural integration of visual and semantic information during perception. These results confirm the complementary nature of visual and textual information streams [79], revealing a synergistic "1+1 > 2" effect that aligns with the brain's use of semantic priors to enrich visual understanding.

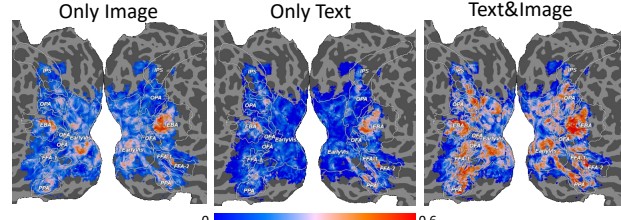

*Figure 7.* Illustrating synergy in the neural encoding task.

**Inter-task Synergy.** A corresponding synergy is observed among decoding tasks, where joint decoding (B → I&T) markedly improves both image and caption outputs over their single-task counterparts (Tabs. 2, 3). Qualitatively, this joint approach exhibits a dual benefit (Fig. 8): it enriches captions with visual details (e.g., color, quantity) inferred from the concurrent image stream, while simultaneously suppressing decoding artifacts. These results collectively validate the benefits of a unified multi-task paradigm for neural decoding.

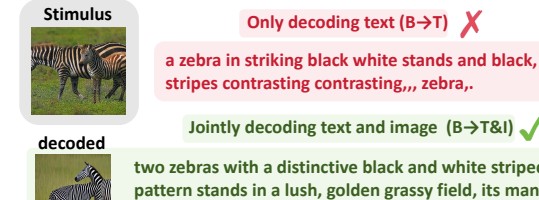

*Figure 8.* Illustrating synergy in the detailed description task.

## 7. Conclusion

This work marks a departure from the prevailing paradigm of specialized models by presenting Mind-Omni, the first single framework to successfully unify seven distinct neural encoding and decoding tasks. Our experiments reveal a critical finding: the unified approach unlocks synergistic gains, particularly in joint-modality tasks, enabling performance that is not only competitive with, but on key semantic and reasoning tasks, superior to larger specialized models. Beyond this demonstration, we distill the architectural and training principles that made this unification possible, offering a concrete blueprint for future research. Moreover, its effectiveness as a computational testbed for neuroscientific exploration highlights the potential of such unified frameworks to advance the holistic modeling of neural cognition.

## Impact Statement

The pursuit of unraveling and emulating the brain's intricate visual processing systems has been a cornerstone endeavor for researchers in computational neuroscience and artificial intelligence. Recent advancements in neural encoding and decoding have opened up numerous possibilities, fueling concerns about the potential harmful use cases of mind reading.

We argue that these concerns can be alleviated for two main reasons: (1) Mind reading requires brain activity recording devices with very high spatial resolution, and data acquisition systems like fMRI, which possess high spatial resolution, are not easily portable; (2) Although there are now several portable brain activity recording devices, achieving mind reading would require the subject to maintain intense focus and cooperate with the data collection process.

## Acknowledgments

This work was supported in part by the National Key R&D Program of China (2023YFF1203501); in part by the National Natural Science Foundation of China under Grant U2441253, 62576336 and 82272072; and in part by the Strategic Priority Research Program of the Chinese Academy of Sciences (XDB0930000).

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

# A. Additional Related Work

## A.1. Unified Models: Autoregressive vs Diffusion

Diffusion models were first introduced for visual content generation, exemplified by models like Stable Diffusion [81]. Subsequently, frameworks such as LLaDa [82] and D3PM [83] began to explore the use of discrete diffusion for modeling language tasks. This line of research recently culminated in models like Mmada [62] and Muddit [39], which successfully unified bi-directional image-text understanding and generation within a single diffusion-based framework, thereby achieving a grand unification of these two modalities. Concurrently, autoregressive (AR) models, with the Transformer [84] as their backbone, have followed a parallel evolutionary path. Originally dominant in language modeling, their capacity for high-fidelity visual generation was demonstrated by frameworks like VQGAN [65] and LlamaGen [85]. More recently, prominent MLLMs including the Emu series [36,37,38], Seed-X [60], and Chameleon [61] have successfully employed the autoregressive paradigm to unify bi-directional image-text tasks, marking a similar milestone for AR-based unification. Additionally, hybrid architectures combining both AR and diffusion principles, such as Show-O [63] and Transfusion [64], have also emerged as a powerful third approach.

The choice between these paradigms involves a critical trade-off, particularly for the unique challenges of neural encoding and decoding. Autoregressive models are highly promising, benefiting from the remarkable scaling laws of large language models. However, their fundamental reliance on a fixed, sequential generation process imposes an artificial causal structure between tasks. This not only complicates data governance as modalities increase, but more critically, it introduces a **confounding bias that obscures the true synergistic relationships** we aim to investigate.

In contrast, the discrete diffusion paradigm's objective of predicting randomly masked tokens makes it largely insensitive to modality ordering. This architectural flexibility provides a crucial advantage: it offers an **unbiased testbed to investigate multi-task synergies concurrently**, without the sequential constraints imposed by AR models. This same flexibility also facilitates a more straightforward extension to the multitude of potential neural modalities (e.g., fMRI, EEG, MEG), each typically associated with scarce datasets. While the performance of current diffusion-based models may trail their AR counterparts in some domains, their architectural suitability for both synergy analysis and future scalability is a decisive advantage for our problem space.

Given these considerations, we selected discrete diffusion as our foundational framework. It presents the most robust and principled approach not only for a **comprehensive investigation into the tri-modal synergies** of the brain-vision-language space, but also as a **scalable pathway** toward a true foundation model capable of seamlessly integrating a broad spectrum of neural data types.

# B. Preliminaries: Discrete Diffusion Modeling

Discrete diffusion models provide a principled framework for generating discrete data, such as text or quantized image tokens. The core idea is to conceptualize a data sample $x$ from a finite alphabet $\mathcal{X} = \{1, \dots, N\}$ as a one-hot vector $\mathbf{x} \in \{0, 1\}^N$. The diffusion process involves progressively corrupting an initial data sample $\mathbf{x}_0$ with noise until it converges to a simple prior distribution, typically a uniform categorical distribution. A generative model is then trained to reverse this process, learning to denoise the corrupted sample and recover the original data.

Following recent formulations [67], this corruption can be modeled as a continuous-time Markov chain (CTMC). We adopt the absorbing-state variant, which has proven highly effective for generative tasks. In this setup, every token can transition to a special absorbing [MASK] token, denoted as $\mathbf{m}$, but can never transition out of it.

**Forward Process.** The forward process describes how the distribution of a token, $p_t$, evolves over time $t \in [0, 1]$ from the initial data distribution $p_0$ to a stationary noise distribution $p_1$. This evolution is governed by the differential equation $\frac{dp_t}{dt} = Q_t\, p_t$, where $Q_t$ is the time-dependent transition rate matrix.

For the absorbing-state model, the posterior probability of a token $x_t$ at time $t$, given the original clean token $\mathbf{x}_0$, is a simple categorical distribution:

$$q(x_t \mid \mathbf{x}_0) = \mathrm{Cat}\big(x_t \mid \alpha_t \mathbf{x}_0 + (1 - \alpha_t)\mathbf{m}\big). \tag{10}$$

Here, $\alpha_t \in [0, 1]$ is the *signal rate* or *survival probability*, representing the chance that a token has not yet been masked by time $t$. Thus, $x_t$ is the original token $\mathbf{x}_0$ with probability $\alpha_t$ and the [MASK] token $\mathbf{m}$ with probability $1 - \alpha_t$.

**Reverse Process.** The generative model learns to approximate the reverse process, which denoises a corrupted token. For any two time points $0 < s < t < 1$, the true reverse posterior can be expressed analytically. This is the distribution we aim to model:

$$q(x_s|x_t, \mathbf{x}_0) = \begin{cases} \mathrm{Cat}(x_s | \frac{(1-\alpha_s)\mathbf{m}+(\alpha_s-\alpha_t)\mathbf{x}_0}{1-\alpha_t}), & \text{if } x_t = \mathbf{m}, \\ \mathrm{Cat}(x_s|x_t), & \text{otherwise.} \end{cases} \tag{11}$$

If a token $x_t$ is not masked, it is preserved when moving backward in time (from $t$ to $s$). If it is masked, its distribution becomes a weighted mixture of the [MASK] token and the original clean token $\mathbf{x}_0$.

**Training Objective.** The model, a network $x_\theta$, is trained to predict the clean token $\mathbf{x}_0$ from a corrupted input $x_t$ and its corresponding time $t$. This is typically achieved by minimizing the continuous-time negative ELBO, which simplifies to a time-weighted cross-entropy loss. Let $\hat{\mathbf{x}}_0 = x_\theta(x_t, t)$ be the model's predicted probability distribution for the clean token. The objective is:

$$\mathcal{L}_{\mathrm{NELBO}} = \mathbb{E}_{t, q(x_t|\mathbf{x}_0)} \left[ w(t) \cdot \left( -\log(\hat{\mathbf{x}}_0 \cdot \mathbf{x}_0) \right) \right], \tag{12}$$

where the weighting function $w(t) = -\alpha'_t/(1 - \alpha_t)$ emphasizes different timesteps during training. This formulation provides a unified foundation for both understanding and generation. Because the corruption schedule and objective are agnostic to the specific semantics of the discrete alphabet $\mathcal{X}$, the same diffusion backbone can naturally unify generation across diverse modalities like text and images.

## C. Data preprocessing

### C.1. NSD Dataset Overview.

We utilize the Natural Scenes Dataset (NSD) [70], a large-scale, high-resolution 7-Tesla fMRI dataset. The dataset comprises neural responses from eight healthy subjects tasked with viewing thousands of natural images sourced from the MS-COCO dataset [86]. Over the course of 30 to 40 sessions, each participant was presented with 9,000–10,000 unique images, each displayed three times for three seconds, yielding a comprehensive set of 22,000–30,000 single-trial fMRI responses per subject. The fMRI data, processed with the GLMSingle methodology [87], consists of session-wise z-scored single-trial beta estimates. Consistent with established protocols in the field [18], our experiments are conducted on the four subjects (subj01, subj02, subj05, and subj07) who completed the full experimental protocol.

### C.2. Multi-subject Data Registration.

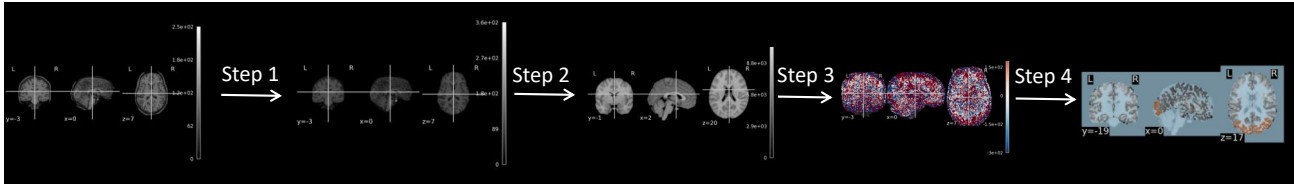

*Figure 9.* Overview of the fMRI registration process.

To address the structural idiosyncrasies and resulting disparate input dimensions across subjects without increasing model parameters, we register the functional data of each participant to the MNI152 [33,34,35] canonical space using FSL [32]. This registration pipeline, illustrated in Fig. 9, involves: (1) skull-stripping the subject-specific T1-weighted data (registered to the functional space: func1pt8mm/T1_to_func1pt8mm.nii.gz), (2) registering this structural scan to the MNI152 template, (3) transforming the functional MRI data into this standard space, and finally, (4) extracting the voxels corresponding to the visual cortex (including early visual areas V1, V2, V3, V4, and higher-order regions such as OPA, OFA, PPA, FFA, EBA, and IPS).

To validate the fidelity of our fMRI registration pipeline, we performed a Representational Similarity Analysis (RSA), as depicted in Fig. 10. For the test set of subject 1, we computed Representational Dissimilarity Matrices (RDMs) from three sources: the fMRI data in its native subject space (left), the same data after registration to the MNI152 space (right), and the CLIP image features of the corresponding stimuli (middle). By correlating the upper triangulars of these matrices, we observe that the registered data preserves the global representational geometry of the original data and its correspondence to the stimulus features. This result confirms the effectiveness of our registration process.

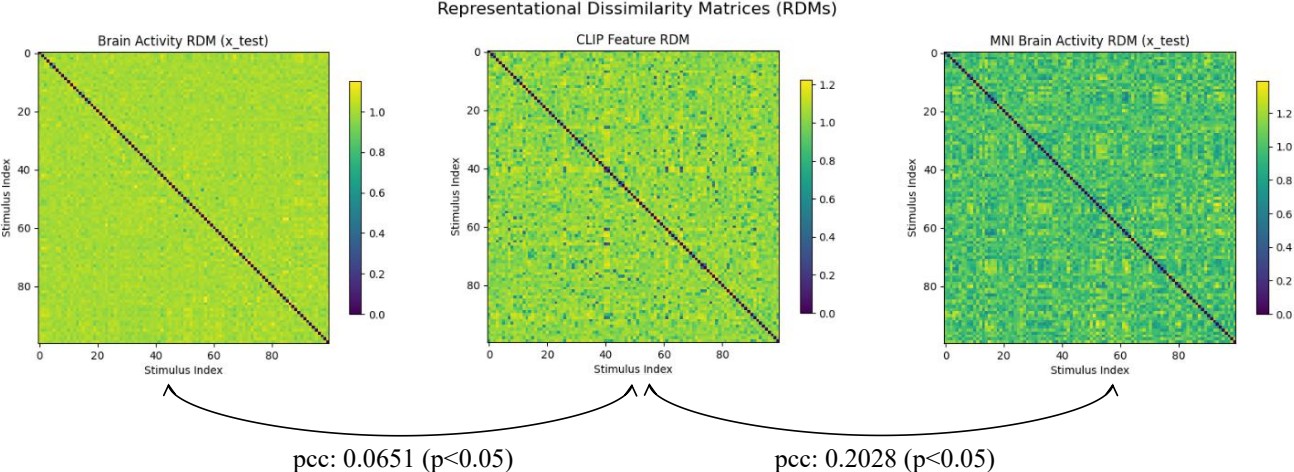

*Figure 10.* Validation of data registration using RDMs. The figure compares Representational Dissimilarity Matrices (RDMs) computed from three sources: (left) the fMRI data in its native subject space (for observation, only the RDMs of the first 100 samples are shown.), (middle) the CLIP visual features of the corresponding image stimuli, and (right) the fMRI data after registration to the MNI152 standard space.

### C.3. MLLM-based Data Curation Details

The original MS-COCO captions often lack the granularity required for high-fidelity semantic reconstruction [56]. To address this, we developed a data curation pipeline utilizing advanced MLLMs, specifically Qwen2-VL (7B) [43] and the LLaVA-Instruct-150K dataset [44,45]. This pipeline synthesizes diverse training samples across three key categories: (1) **Caption Enrichment**, transforming brief labels into dense, attribute-rich descriptions; (2) **Multi-Granularity Instructions**, enabling the model to adaptively generate either detailed or concise outputs based on varied user prompts; and (3) **Reasoning Q&A**, incorporating visual reasoning and question-answering tasks to facilitate deeper semantic understanding. Specific methodologies are detailed below.

**1. Caption Enrichment.** To generate high-quality, dense captions, we fed both the raw stimulus images and their original COCO captions into Qwen2-VL. We instructed the model to expand the descriptions while strictly preserving core semantic fidelity. These enriched captions constitute the textual component of our paired brain-image-text data, serving as the foundation for the Brain Captioning task. The specific prompt used is as follows:

---

**Prompt: Caption Data Enrichment**

```
Please rewrite the provided COCO caption into a slightly more detailed image
description, but never longer than 40 words.  Keep your sentences to a maximum of 40
words.  When rewriting, please consider the following points:
1. Specify Objects:  Provide detailed descriptions of the main objects in the image,
   including their color, material, condition, and actions.
2. Maintain Natural Flow:  Ensure the rewritten description is grammatically correct
   and expresses ideas fluently.
3. Keep your sentences to a maximum of 40 words.
```

---

**2. Multi-Granularity Instructions.** To bolster the model's instruction-following capabilities, we curated datasets for both *detailed* and *concise* generation tasks.

*Detailed Descriptions:* While the initial enrichment step successfully expanded visual details, the resulting captions exhibited significant variance in syntactic structure and length. To standardize the data distribution and ensure a consistent, canonical response format for the detailed description task, we employed a few-shot prompting strategy to further refine these captions. Acting as an "expert editor," the model was guided by three manually crafted examples to distill the enriched texts into clear and structurally uniform paragraphs (approx. 30 words). The prompt template is detailed below:

---

**Prompt: Detailed Description Rephrasing (Few-Shot)**

```
Your task is to act as an expert editor.  Rephrase the following sentences to be clear,
concise, and around 30 words, while strictly preserving the original meaning.  Follow
the approach demonstrated in the three examples provided below.
Example 1:
Original:  At a bustling outdoor market, a man in a white shirt and green cap sorts
through a large pile of vibrant orange carrots, surrounded by bags of potatoes and
other fresh produce.
Rephrased:  The image depicts a bustling open-air market filled with people shopping
for vegetables.  Large piles of carrots and potatoes are prominently displayed
throughout the market, drawing the shoppers' attention.  At least sixteen people can
be seen browsing, interacting, and shopping in the market area, some very close to the
vegetable piles.
Example 2:
Original:  The image is a well-lit display case filled with an array of cakes...  [Full
example omitted]
Rephrased:  The image showcases a beautifully illuminated display case in a room,
exhibiting a variety of cakes on several shelves...  [Full example omitted]
Example 3:
Original:  Three friends sit at a wooden table...  [Full example omitted]
Rephrased:  The image shows a scene that three friends are sitting...  [Full example
omitted]
Now, rephrase this text:  {raw_caption}
```

To ensure robustness against diverse user queries, we paired these detailed responses with instructions randomly sampled from the following pool:

---

**Instruction Pool: Detailed Description Queries**

```
• "What do you see happening in this image?"
• "What do you think is going on in this snapshot?"
• "Can you elaborate on the elements of the picture provided?"
• "Describe the following image."
• "Write a detailed description of the given image."
• "Explain the visual content of the image in great detail."
• "Analyze the image in a comprehensive and detailed manner."
• "Can you describe the main features of this image for me?"
• "Describe the following scene."
• "What are the key elements in this picture?"
```

*Concise Descriptions:* For the concise description task, we utilized the original MS-COCO captions as ground truth targets. To train the model to respond to requests for brevity, we paired these captions with a diverse set of instructions, such as *"Give me a very very short description of the scene," "Please briefly describe the content of the picture," "Describe the picture's content with concise language,"* and *"Please inform me of the picture's content briefly."*

**3. Reasoning Q&A.** To instill reasoning abilities, we retrieved Question-Answer (QA) pairs from LLaVA-Instruct-150K by matching the COCO IDs of our stimulus images. Since the original answers often exceed our token limits, we utilized an MLLM to condense the answers to a maximum of 77 tokens using the following summarization prompt:

---

**Prompt: Reasoning Q&A Summarization**

```
You are a professional text summarizer.  Your task is to rephrase and condense the
given Answer of a Q&A pair to approximately 30 words.  You must preserve the core
meaning, key details, and original tone.  Remove any redundant phrases or unnecessary
elaborations.
Original reasoning Q&A: Question: {Q}.  Answer: {A}
```

*Manual Verification:* A critical challenge with LLaVA-Instruct-150K is the presence of questions relying heavily on external commonsense rather than immediate visual evidence (e.g., asking about *"health risks for cows"* or *"safety factors for plane landings"*). Such questions are ill-suited for our task, which focuses on decoding visual perception from brain activity. **To address this, we conducted a rigorous manual verification process (approximately 20 person-hours) to filter out**

**instances with weak visual grounding. This ensures that the reasoning dataset is strictly aligned with the visual content presented in the stimuli.**

Examples from the curated dataset are shown in Fig. 11. An overview of the datasets used for model training is provided in Tab. 7. We will release all processed fMRI data and the instruction-tuning datasets to the community to facilitate further research.

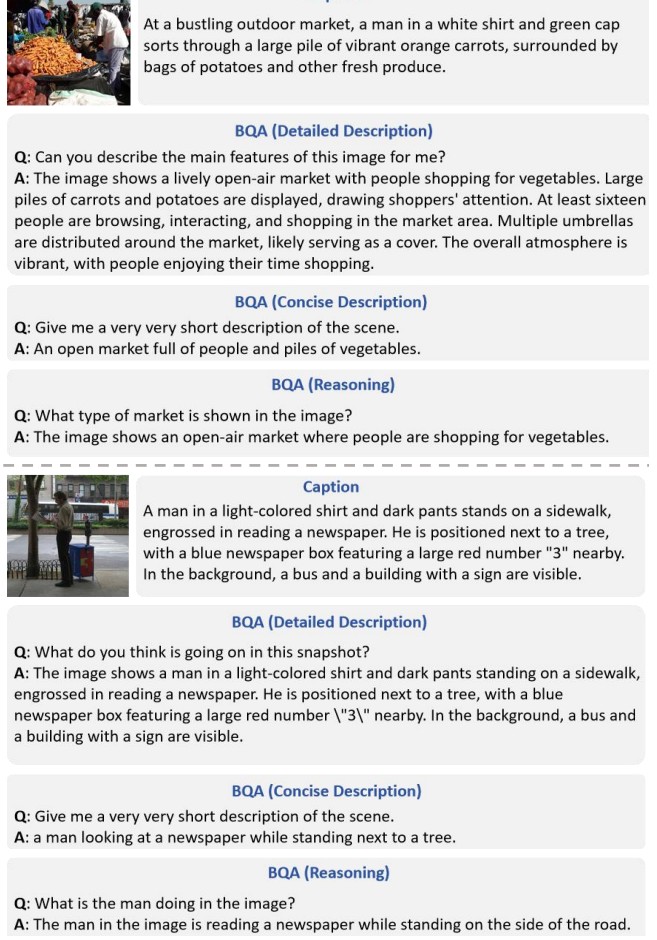

*Figure 11.* Examples of MLLM-curated image-text pairs. Each stimulus image is associated with two types of textual data: a paired caption, and a set of BQA question-answer pairs that include concise descriptions, detailed descriptions, and reasoning tasks.

*Table 7.* Overview of the dataset composition. We distinguish between the number of unique visual-linguistic stimuli (Instruction-Response pairs) and the corresponding fMRI data volumes. Each stimulus is associated with multiple single-trial fMRI responses (typically 3 repetitions) and one averaged multi-trial response.

| | Stimulus Scale | fMRI Data Volume | |
|---|---|---|---|
| **Task / Data Type** | # Unique Pairs | Single-Trial | Multi-Trial (Avg) |
| ***Basic Alignment*** | | | |
| Image-Caption paired data | 72K | 216K | 72K |
| ***Instruction Tuning*** | | | |
| Detailed Description | 72K | 216K | 72K |
| Concise Description | 72K | 216K | 72K |
| Reasoning Q&A | 58K | 174K | 58K |
| **Total Samples** | **274K** | **822K** | **274K** |

# D. Hyperparameter Configuration

**fMRI Predictor.**   Prior to training the Brain Tokenizer, we first train a 4-layer MLP with residual connections to serve as the fMRI predictor, $P_{\text{fMRI}}$. This model is trained on the single-trial data from all 8 subjects to map fMRI signals to the 1024-dimensional image and text feature space. The training hyperparameters are: a batch size of 4096, a learning rate of 5e-4, and an 8-bit Adam optimizer ($\beta_1$=0.9, $\beta_2$=0.999). The model was trained for 40 epochs on a single A100 GPU.

**Brain Tokenizer.**   The Brain Tokenizer is trained on the single-trial data from all 8 subjects with the following configuration: a codebook size of 128, a code dimension of 16, and a total of 64 codes per fMRI sample. The loss weights are set to $\lambda = 0.5$, $\lambda_1 = 0.08$, $\lambda_2 = 0.02$, and the commitment loss weight $\beta = 0.8$. The codebook is updated via Exponential Moving Average (EMA) with a decay of 0.99. We use a batch size of 128 with 4 gradient accumulation steps. The model is trained for 14,000 steps on four A100 GPUs using an 8-bit Adam optimizer ($\beta_1$=0.9, $\beta_2$=0.999) with a constant learning rate of 2e-4, following a 300-step warmup. Throughout this stage, the parameters of the fMRI predictor are kept frozen.

**MM-DiT Backbone.**   The training of the backbone is divided into two stages, employing a progressive curriculum that moves from simpler to more complex tasks with varying data mixing ratios. The detailed hyperparameter configurations for each stage are provided in Table 8. Notably, while the [MASK] tokens for the image and text modalities are inherited from Muddit, the brain [MASK] token is randomly initialized, and all three are kept frozen throughout training.

*Table 8.* Detailed hyperparameter configurations used across the various training stages.

| Training Stage | | Stage 1 | | Stage 2 |
| --- | --- | --- | --- | --- |
| | | Stage 1.1 | Stage 1.2 | |
| Method | Backbone | Frozen | Frozen | DoRA (r=8, alpha=16) |
| | New Module | Trainable | Trainable | Trainable |
| Task Allocation | | I+T→B, B→I+T | I+T→B, I→B, T→B, B→I+T, B→I, B→T | I+T→B, B→I+T, BQA |
| Data | | Single trial | Single trial | Mixed |
| Data Mixing Ratio | | 1:1 | 1:2:2:1:2:2 | 1:1:2 |
| Learning Rate | | 5e-5 (constant) | 5e-5 (constant) | 5e-5 (Cosine decay) |
| Warmup Steps | | 300 | 300 | 0 |
| Batch Size | | 96 | 96 | 32 |
| Accumulation Steps | | 3 | 3 | 8 |
| Mixed Precision | | bf16 | bf16 | bf16 |
| Optimizer | | 8-bit Adam | 8-bit Adam | 8-bit Adam |
| Betas | | $\beta_1$=0.9, $\beta_2$=0.999 | $\beta_1$=0.9, $\beta_2$=0.999 | $\beta_1$=0.9, $\beta_2$=0.999 |
| Weight decay | | 1e-2 | 1e-2 | 1e-2 |
| Training Steps | | 16500 | 7500 | 1800 |
| Computational resources | | 2 A100 | 2 A100 | 4 A100 |

# E. More Decoding Results

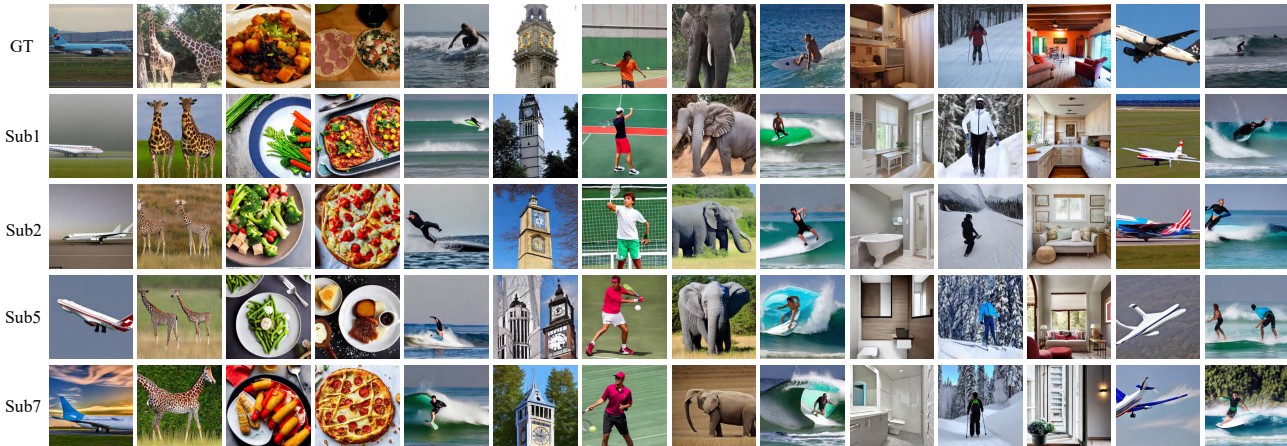

*Figure 12.* More cross-subject reconstructions of Mind-Omni on subject 1, 2, 5 and 7.

*Table 9.* Per-subject performance of our model on concise description, detailed description, and reasoning tasks.

| Subject | Method | BLEU1↑ | BLEU2↑ | BLEU3↑ | BLEU4↑ | METEOR↑ | ROUGE↑ | CIDEr↑ | SPICE↑ | CLIP-S↑ | RefCLIP↑ | BERT↑ |
|---|---|---|---|---|---|---|---|---|---|---|---|---|
| | | | | | **Concise Description** | | | | | | | |
| sub1 | B→T | 15.75 | 5.11 | 1.47 | 0.50 | 14.72 | 19.26 | 5.70 | 4.33 | 56.66 | 60.97 | 82.37 |
| | B→I&T | 30.07 | 15.50 | 8.26 | 4.71 | 22.14 | 25.45 | 12.55 | 9.24 | 52.47 | 51.02 | 87.96 |
| sub2 | B→T | 13.39 | 4.72 | 1.22 | 0.34 | 15.10 | 19.18 | 5.88 | 4.34 | 57.18 | 61.36 | 81.44 |
| | B→I&T | 30.08 | 15.74 | 8.60 | 5.08 | 22.30 | 25.76 | 14.23 | 9.12 | 53.37 | 51.47 | 87.97 |
| sub5 | B→T | 13.52 | 4.94 | 1.13 | 0.49 | 15.23 | 19.70 | 6.51 | 5.00 | 58.49 | 62.08 | 81.49 |
| | B→I,T | 31.02 | 16.58 | 9.16 | 5.37 | 23.11 | 26.47 | 15.46 | 10.53 | 55.25 | 53.81 | 88.23 |
| sub7 | B→T | 13.27 | 4.27 | 1.01 | 0.34 | 14.83 | 19.15 | 6.16 | 3.59 | 56.23 | 61.78 | 81.17 |
| | B→I&T | 30.29 | 15.55 | 8.33 | 4.84 | 22.16 | 25.17 | 12.59 | 8.55 | 51.43 | 50.07 | 87.82 |
| | | | | | **Detail Description** | | | | | | | |
| sub1 | B→T | 14.30 | 8.62 | 5.53 | 3.66 | 12.82 | 15.04 | 5.42 | 6.34 | 46.23 | 45.77 | 78.84 |
| | B→I&T | 28.60 | 17.23 | 11.05 | 7.31 | 25.63 | 30.07 | 10.84 | 12.68 | 52.46 | 51.53 | 87.68 |
| sub2 | B→T | 15.27 | 9.19 | 5.89 | 3.87 | 13.48 | 15.81 | 6.20 | 6.86 | 47.88 | 47.33 | 81.59 |
| | B→I&T | 29.36 | 17.67 | 11.32 | 7.44 | 25.92 | 30.40 | 11.93 | 13.19 | 53.62 | 52.55 | 87.68 |
| sub5 | B→T | 14.23 | 8.68 | 5.64 | 3.74 | 12.81 | 15.05 | 6.54 | 6.96 | 51.08 | 46.67 | 82.23 |
| | B→I&T | 29.65 | 18.09 | 11.74 | 7.79 | 26.69 | 31.35 | 13.63 | 14.50 | 56.42 | 55.57 | 87.97 |
| sub7 | B→T | 15.87 | 9.63 | 6.24 | 4.15 | 14.27 | 16.70 | 6.95 | 6.95 | 48.70 | 48.23 | 78.17 |
| | B→I&T | 28.86 | 17.51 | 11.34 | 7.55 | 25.95 | 30.36 | 12.64 | 12.64 | 52.18 | 51.33 | 87.59 |
| | | | | | **Reasoning** | | | | | | | |
| sub1 | BQA | 23.32 | 15.98 | 11.99 | 9.44 | 50.38 | 53.10 | 227.70 | 43.27 | 70.88 | 76.89 | 81.94 |
| sub2 | BQA | 23.16 | 15.80 | 11.86 | 9.33 | 49.97 | 52.92 | 227.54 | 42.99 | 70.32 | 76.46 | 81.97 |
| sub5 | BQA | 23.28 | 15.88 | 11.86 | 9.31 | 50.45 | 53.21 | 221.93 | 43.48 | 70.90 | 76.99 | 81.99 |
| sub7 | BQA | 22.96 | 15.66 | 11.74 | 9.23 | 49.72 | 52.41 | 218.39 | 43.38 | 70.50 | 76.55 | 81.96 |

*Table 10.* Per-subject performance of our model on the decoding task. We evaluate metrics at both the low and high semantic levels.

| Subject | Method | Low-Level | | | | High-Level | | | |
|---|---|---|---|---|---|---|---|---|---|
| | | PixCorr ↑ | SSIM ↑ | AlexNet(2) ↑ | AlexNet(5) ↑ | Inception ↑ | CLIP ↑ | EffNet-B ↓ | SwAV ↓ |
| sub1 | B→T | 0.033 | 0.284 | 0.629 | 0.706 | 0.661 | 0.691 | 0.903 | 0.631 |
| | B→I&T | 0.058 | 0.341 | 0.762 | 0.865 | 0.778 | 0.784 | 0.823 | 0.532 |
| sub2 | B→T | 0.031 | 0.283 | 0.628 | 0.715 | 0.680 | 0.698 | 0.898 | 0.619 |
| | B→I&T | 0.057 | 0.343 | 0.712 | 0.843 | 0.793 | 0.813 | 0.824 | 0.538 |
| sub5 | B→T | 0.044 | 0.286 | 0.636 | 0.714 | 0.715 | 0.732 | 0.875 | 0.609 |
| | B→I&T | 0.065 | 0.339 | 0.723 | 0.862 | 0.814 | 0.826 | 0.806 | 0.523 |
| sub7 | B→T | 0.034 | 0.282 | 0.621 | 0.680 | 0.660 | 0.674 | 0.906 | 0.636 |
| | B→I&T | 0.051 | 0.342 | 0.703 | 0.825 | 0.768 | 0.772 | 0.843 | 0.558 |

We present the reconstruction results for multiple subjects in Fig. 12, demonstrating the consistency of our model's performance across individuals.

Figure 13 presents qualitative results for various language decoding sub-tasks. The results demonstrate that our multi-task framework, Mind-Omni, can not only follow instructions to generate both concise and detailed descriptions but also execute reasoning tasks based on the visual stimulus. This highlights the versatility and generalizability of our model.

Finally, detailed quantitative results for the image and text decoding tasks across multiple subjects are provided in Tab. 10 and Tab. 9.

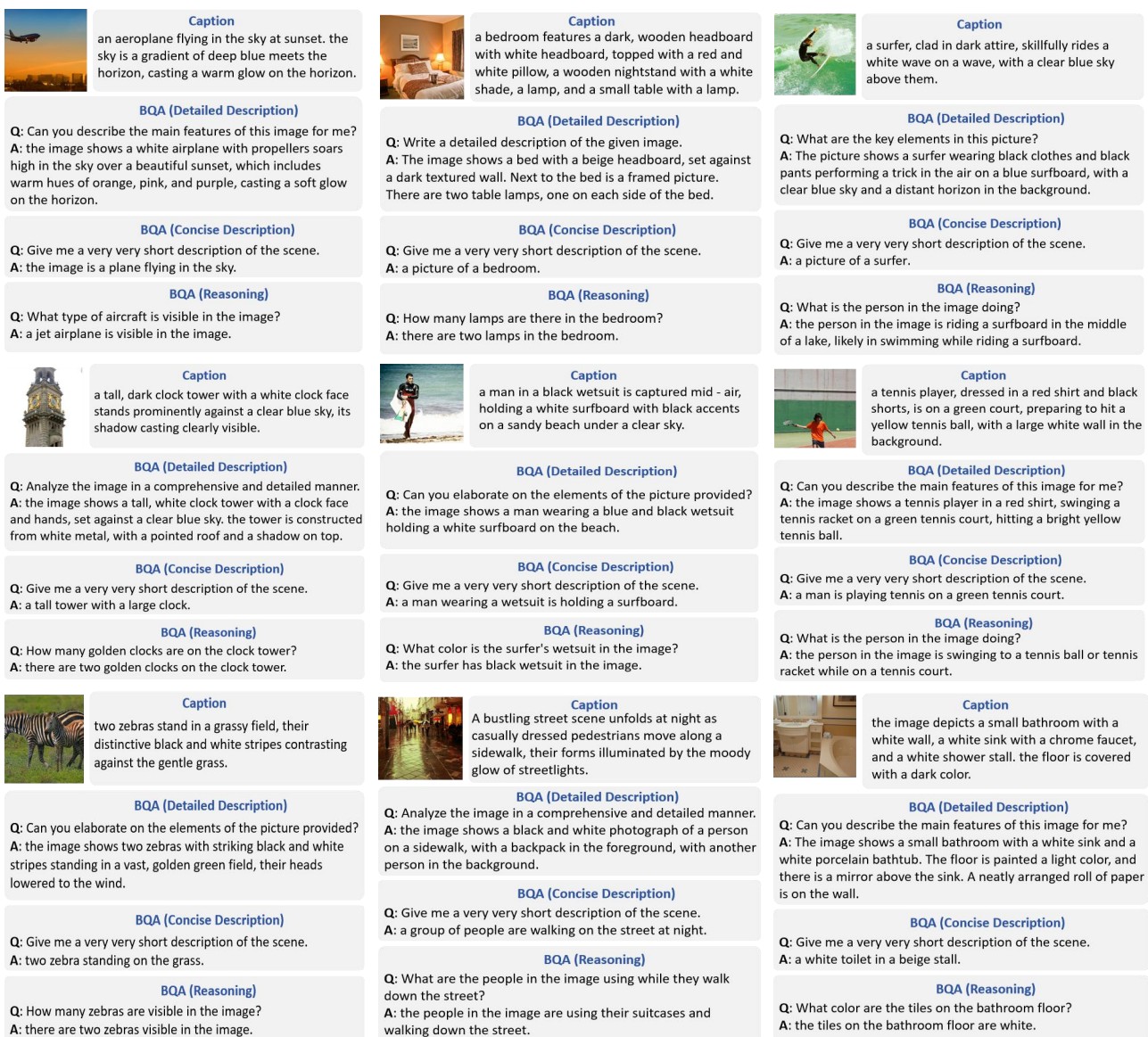

*Figure 13.* Additional qualitative text decoding results for Subject 1. The figure illustrates performance across diverse tasks: Brain Captioning, concise and detailed descriptions, and reasoning.

## F. More Encoding Results

We provide a detailed breakdown of the per-subject neural encoding results in Table 11, with corresponding visualizations for the bi-modal encoding task presented in Fig. 14.

*Table 11.* Per-subject performance of our model on neural encoding task. We evaluate metrics at both the voxel level (rPCC, rMSE, rRSA) and the semantic level ($PCC_{perceptual}$, $MSE_{perceptual}$, $RSA_{perceptual}$). Arrows indicate the preferred direction for each metric (↑ higher is better, ↓ lower is better).

| Subject | Method | Voxel-Level | | | Semantic-Level | | |
|---|---|---|---|---|---|---|---|
| | | gPCC↑ | gMSE↓ | gRSA↑ | $PCC_{semantic}$↑ | $MSE_{semantic}$↓ | $RSA_{semantic}$↑ |
| sub1 | T→B | 0.103 | 0.740 | 0.296 | 0.739 | 0.198 | 0.619 |
| | I→B | 0.159 | 0.662 | 0.409 | 0.679 | 0.225 | 0.563 |
| | I&T→B | 0.156 | 0.666 | 0.411 | 0.754 | 0.187 | 0.654 |
| sub2 | T→B | 0.116 | 0.727 | 0.317 | 0.739 | 0.198 | 0.619 |
| | I→B | 0.171 | 0.654 | 0.422 | 0.679 | 0.225 | 0.563 |
| | I&T→B | 0.178 | 0.646 | 0.430 | 0.754 | 0.187 | 0.654 |
| sub5 | T→B | 0.126 | 0.729 | 0.353 | 0.739 | 0.198 | 0.619 |
| | I→B | 0.175 | 0.653 | 0.457 | 0.679 | 0.225 | 0.563 |
| | I&T→B | 0.180 | 0.647 | 0.455 | 0.754 | 0.187 | 0.654 |
| sub7 | T→B | 0.085 | 0.728 | 0.108 | 0.739 | 0.198 | 0.619 |
| | I→B | 0.122 | 0.656 | 0.322 | 0.679 | 0.225 | 0.563 |
| | I&T→B | 0.124 | 0.656 | 0.334 | 0.754 | 0.187 | 0.654 |

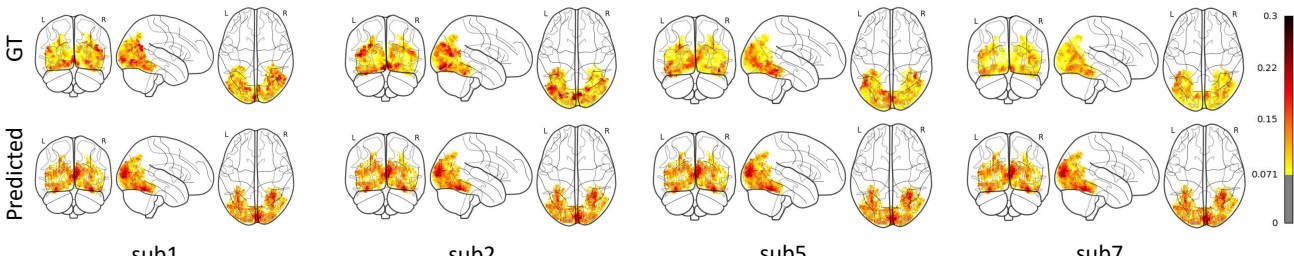

*Figure 14.* Visualization of bi-modal (image and text) neural encoding on subjects 1, 2, 5, and 7, with results averaged over 1000 test samples.

As shown in Tab. 4, our model surpasses previous state-of-the-art (SOTA) models on all semantic-level metrics for neural encoding. However, its performance on voxel-level metrics lags significantly behind. We attribute this discrepancy to two factors. First, the Brain Tokenizer performs a lossy compression of fMRI signals, which may lead to suboptimal fMRI generation. Second, our model employs a VQ-VAE [65] for image features and per-token CLIP [66] embeddings for text, whereas prior SOTA models typically use pooled features from the final layer of CLIP. We found that even a simple linear regression model can achieve results comparable to the SOTA (MindSimulator [25]) when using the same pooled CLIP features. Therefore, we hypothesize that the underperformance of our Mind-Omni model stems from the insufficient neural plausibility of the image and text encoders within its foundation model, Muddit [39].

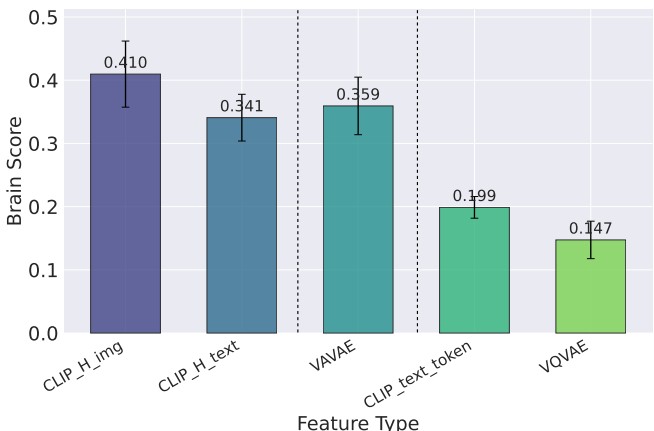

*Figure 15.* Comparative brain-like performance (Brain Score [88]) of different image and text encoders.

To validate this hypothesis, we evaluated the brain-alignment of several feature extractors using Brain Score [88]: (1) understanding-oriented encoders: CLIP_H_img and CLIP_H_text; (2) generation-oriented encoders (as used in our model): VQ-VAE and per-token CLIP embeddings; and (3) a recently proposed unified model: VAVAE [89]. As illustrated in Fig. 15, the understanding-oriented models achieve the highest Brain Scores, consistent with the choices in prior SOTA neural encoding works. In contrast, our model's encoders exhibit lower Brain Scores. Intriguingly, the unified VAVAE model also demonstrates a high Brain Score.

These findings suggest a trade-off: understanding-specialized models excel at neural encoding but are unsuitable for decoding, while generation-specialized models are effective for decoding but perform poorly on encoding. This also highlights a core challenge for unified models. A promising solution lies in unified architectures like VAVAE [89] or RAE [90]. In future work, we will explore such models to simultaneously enhance both encoding and decoding performance.

## G. Additional Ablation Studies

In Tab. 12, we fix the architectural parameters of the Brain Tokenizer and conduct a detailed ablation study on its various loss components. The results indicate that $\mathcal{L}_{\textbf{coarse-grain}}$ is the most critical component, contributing predominantly to the semantic alignment of the brain tokens. In contrast, $\mathcal{L}_{\textbf{fine-grain}}$ and $\mathcal{L}_{\textbf{perceptual}}$ primarily enhance the diversity and richness of the codebook through fine-grained alignment, thereby increasing the overall codebook usage.

Tables 13–18 provide a detailed supplement to the results presented in Tab. 6 of the main text.

*Table 12.* Ablation study on different loss function combinations for the Brain Tokenizer. rPCC is calculated on self-reconstructed fMRI signals. The chance level for retrieval is 0.05. The blue-shaded row denotes the strategy adopted by our model.

| $\mathcal{L}_{\textbf{coarse-grain}}$ | $\mathcal{L}_{\textbf{fine-grain}}$ | $\mathcal{L}_{\textbf{perceptual}}$ | codebook size | code dim. | code num. | rPCC ↑ | Retrieval (Top50) ↑ | | codebook usage ↑ |
|---|---|---|---|---|---|---|---|---|---|
| | | | | | | | **B2I** | **B2T** | |
| × | × | × | 128 | 16 | 64 | 0.45 | 0.05 | 0.05 | 32% |
| × | × | ✓ | 128 | 16 | 64 | 0.55 | 0.11 | 0.08 | 46% |
| × | ✓ | ✓ | 128 | 16 | 64 | 0.57 | 0.32 | 0.28 | 56% |
| ✓ | × | ✓ | 128 | 16 | 64 | 0.66 | 0.60 | 0.55 | 68% |
| ✓ | ✓ | × | 128 | 16 | 64 | 0.68 | 0.60 | 0.57 | 62% |
| ✓ | ✓ | ✓ | 128 | 16 | 64 | **0.68** | **0.62** | **0.59** | **80%** |

*Table 13.* A supplementary table to Tab. 6(a) on the choice of tokenization strategy. We compare the performance of different strategies (code dim.=32 vs. code dim.=16) on the Concise Description task after stage 1 training. Results are averaged across subjects 1, 2, 5, and 7. The blue-shaded row denotes the strategy adopted by our model.

| Method | BLEU1↑ | BLEU2↑ | BLEU3↑ | BLEU4↑ | METEOR↑ | ROUGE↑ | CIDEr↑ | SPICE↑ | CLIP-S↑ | RefCLIP↑ | BERT↑ |
|---|---|---|---|---|---|---|---|---|---|---|---|
| | | | | | Concise Description | | | | | | |
| code dim.=32, code num.=32 | 29.56 | 15.31 | 8.44 | 4.92 | **22.39** | **25.85** | **14.40** | **9.64** | **54.34** | **52.62** | **88.32** |
| code dim.=16, code num.=64 | **30.28** | **15.47** | **8.56** | **4.99** | 22.27 | 25.73 | 13.68 | 9.35 | 53.83 | 52.28 | 88.04 |

*Table 14.* A supplementary table to Tab. 6(a) on the choice of tokenization strategy. We compare the performance of different strategies (code dim.=32 vs. code dim.=16) on the Neural Encoding task after stage 1 training. Results are averaged across subjects 1, 2, 5, and 7. The blue-shaded row denotes the strategy adopted by our model.

| Method | Voxel-Level | | | Semantic-Level | | |
|---|---|---|---|---|---|---|
| | rPCC↑ | rMSE↓ | rRSA↑ | $PCC_{semantic}$↑ | $MSE_{semantic}$↓ | $RSA_{semantic}$↑ |
| code dim.=32, code num.=32 | 0.126 | **0.621** | 0.315 | 0.741 | 0.188 | 0.625 |
| code dim.=16, code num.=64 | **0.145** | 0.698 | **0.342** | **0.760** | **0.184** | **0.664** |

## H. Cross-Modal Synergy Reveals Implicit Semantic Processing

As quantitatively demonstrated in Tab. 4, joint image-text encoding (I&T→B) consistently outperforms unimodal encoding with either images (I→B) or text (T→B) alone, across both voxel-level and semantic-level metrics. To explore this cross-modal synergy in greater detail, we employ cortical projection visualization[1]. For four subjects (1, 2, 5, and 7), we computed

---

[1] Available at https://github.com/gallantlab/pycortex

*Table 15.* A supplementary table to Tab. 6(b) on the choice of training strategy. We compare the performance of different strategies (Direct vs. Progressive) on the Concise Description, Detail Description, and Reasoning tasks after full training. Results are averaged across subjects 1, 2, 5, and 7. The blue-shaded row denotes the strategy adopted by our model.

| Method | BLEU1↑ | BLEU2↑ | BLEU3↑ | BLEU4↑ | METEOR↑ | ROUGE↑ | CIDEr↑ | SPICE↑ | CLIP-S↑ | RefCLIP↑ | BERT↑ |
|---|---|---|---|---|---|---|---|---|---|---|---|
| | | | | | **Concise Description** | | | | | | |
| Direct | 28.49 | 14.53 | 7.11 | 3.92 | 22.17 | 22.05 | 12.09 | 8.71 | 50.60 | 49.87 | 85.41 |
| Progressive | **30.37** | **15.84** | **8.59** | **5.00** | **22.43** | **25.71** | **13.71** | **9.36** | **53.13** | **51.59** | **88.00** |
| | | | | | **Detail Description** | | | | | | |
| Direct | 29.11 | 16.69 | 10.01 | 6.31 | 24.65 | 27.31 | 11.43 | 11.32 | 51.01 | 51.74 | 84.29 |
| Progressive | **29.12** | **17.63** | **11.36** | **7.52** | **26.05** | **30.54** | **12.26** | **13.25** | **53.67** | **52.75** | **87.73** |
| | | | | | **Reasoning** | | | | | | |
| Direct | 22.36 | 13.74 | 10.54 | 9.08 | 46.97 | 51.78 | 214.61 | 43.12 | 67.45 | 74.43 | 80.70 |
| Progressive | **23.18** | **15.83** | **11.86** | **9.33** | **50.13** | **52.91** | **223.98** | **43.28** | **70.65** | **76.72** | **81.96** |

*Table 16.* A supplementary table to Tab. 6(b) on the choice of training strategy. We compare the performance of different strategies (Direct vs. Progressive) on the Neural Encoding task after full training. Results are averaged across subjects 1, 2, 5, and 7. The blue-shaded row denotes the strategy adopted by our model.

| Method | Voxel-Level | | | Semantic-Level | | |
|---|---|---|---|---|---|---|
| | gPCC↑ | gMSE↓ | gRSA↑ | $PCC_{semantic}$↑ | $MSE_{semantic}$↓ | $RSA_{semantic}$↑ |
| Direct | 0.132 | 0.677 | 0.367 | 0.731 | **0.187** | 0.635 |
| Progressive | **0.160** | **0.654** | **0.408** | **0.754** | **0.187** | **0.654** |

*Table 17.* A supplementary table to Tab. 6(c) on the choice of training data. We compare the performance on different data sources (Raw COCO vs. Qwen2-VL Enhanced) on the Concise Description, Detail Description, and Reasoning tasks after full training. Results are averaged across subjects 1, 2, 5, and 7. The blue-shaded row denotes the data source adopted by our model.

| Method | BLEU1↑ | BLEU2↑ | BLEU3↑ | BLEU4↑ | METEOR↑ | ROUGE↑ | CIDEr↑ | SPICE↑ | CLIP-S↑ | RefCLIP↑ | BERT↑ |
|---|---|---|---|---|---|---|---|---|---|---|---|
| | | | | | **Concise Description** | | | | | | |
| Raw COCO | 24.32 | 13.42 | 5.76 | 3.13 | 20.35 | 20.17 | 11.74 | 8.04 | 52.74 | 51.35 | 84.61 |
| Qwen2-VL Enhanced | **30.37** | **15.84** | **8.59** | **5.00** | **22.43** | **25.71** | **13.71** | **9.36** | **53.13** | **51.59** | **88.00** |
| | | | | | **Detail Description** | | | | | | |
| Raw COCO | 24.37 | 12.53 | 8.14 | 3.48 | 23.75 | 26.52 | 9.94 | 10.20 | 49.36 | 50.43 | 83.17 |
| Qwen2-VL Enhanced | **29.12** | **17.63** | **11.36** | **7.52** | **26.05** | **30.54** | **12.26** | **13.25** | **53.67** | **52.75** | **87.73** |
| | | | | | **Reasoning** | | | | | | |
| Raw COCO | 21.42 | 12.38 | 9.65 | 8.42 | 46.36 | 48.56 | 174.35 | 40.53 | 65.61 | 70.58 | 79.62 |
| Qwen2-VL Enhanced | **23.18** | **15.83** | **11.86** | **9.33** | **50.13** | **52.91** | **223.98** | **43.28** | **70.65** | **76.72** | **81.96** |

*Table 18.* A supplementary table to Tab. 6(c) on the choice of training data. We compare the performance on different data sources (Raw COCO vs. Qwen2-VL Enhanced) on the Neural Encoding task after full training. Results are averaged across subjects 1, 2, 5, and 7. The blue-shaded row denotes the data source adopted by our model.

| Method | Voxel-Level | | | Semantic-Level | | |
|---|---|---|---|---|---|---|
| | gPCC↑ | gMSE↓ | gRSA↑ | $PCC_{semantic}$↑ | $MSE_{semantic}$↓ | $RSA_{semantic}$↑ |
| Raw COCO | 0.133 | 0.671 | 0.384 | 0.734 | 0.195 | 0.621 |
| Qwen2-VL Enhanced | **0.160** | **0.654** | **0.408** | **0.754** | **0.187** | **0.654** |

the mean Pearson Correlation Coefficient (PCC) between predicted and ground-truth fMRI responses over 100 randomly selected test samples. These mean PCCs are projected onto the cortical surface, as visualized in Fig. 16, where red indicates higher prediction accuracy (higher PCC) and blue signifies the converse.

The visualizations in Fig. 16 reveal distinct encoding patterns. Image-only encoding is effective in both early (V1, V2, V3) and higher-level (e.g., EBA, PPA) visual areas, whereas text-only encoding is primarily effective in high-level semantic regions (EBA, FFA, PPA). **Strikingly, the joint image-text model achieves high accuracy across the entire visual cortex. This finding is particularly noteworthy given that subjects were only presented with visual stimuli during the fMRI experiment. We posit that this synergy is not merely a computational artifact but rather reflects a fundamental cognitive process: the brain's spontaneous invocation of semantic priors during visual perception [24,26,91].** In

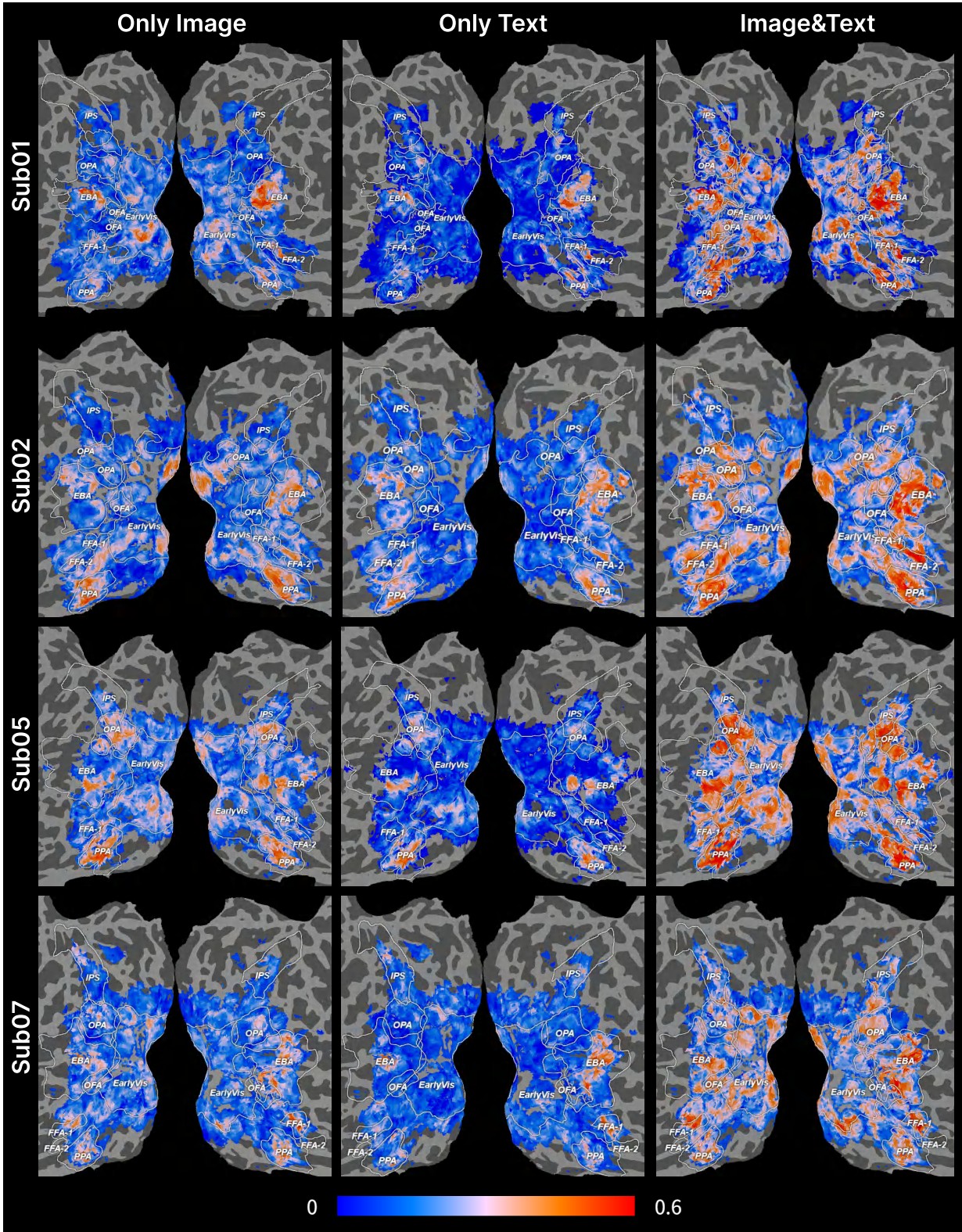

*Figure 16.* Visual comparison of unimodal versus multimodal neural encoding performance. The predicted-to-groundtruth Pearson Correlation Coefficients (PCCs) from the test set are projected onto the cortical surface for four subjects. Red indicates higher PCC values, while blue indicates lower values. See Section H.

essence, when observing a natural image, the human brain likely engages associated linguistic and semantic concepts to enrich comprehension. **The superior performance of the joint model suggests it successfully captures this integrated neural processing, showcasing a synergistic "$1 + 1 > 2$" effect that is consistently observed across all subjects.** This result provides strong evidence that the cross-modal synergy in our model mirrors the brain's natural integration of visual and semantic information.

# I. Mind-Omni as a Computational Testbed: Preliminary Explorations into Conceptual Processing

In this section, we leverage the trained Mind-Omni model as a computational testbed to simulate the brain's responses to external stimuli. We focus on conceptual representation in the human brain as a case study [25], showcasing the potential of our unified model for exploring such frontier scientific questions. Our investigation is structured as follows: We first review the relevant neuroscientific background in Section I.1. Then, in Section I.2, we validate our model's effectiveness by confirming its ability to capture well-established category-selectivity in the visual cortex. Finally, building on this validation, we conduct preliminary explorations of novel concept-selective regions using Mind-Omni in Section I.3.

## I.1. Preliminaries: Category-Selectivity in the Visual Cortex

The human visual system is organized hierarchically [92,93], processing visual information from simple features to complex conceptual representations. The journey begins in the early visual cortex (EarlyVis), encompassing areas like V1, V2, and V3. These regions are primarily responsible for analyzing low-level features of the visual input, such as edges, orientations, spatial frequencies, and colors [94]. As information ascends from the EarlyVis, it enters the higher-level visual cortex, where these basic features are integrated into coherent perceptions of objects, shapes, and scenes [95,96].

While the complete neural code for object recognition is understood to be complex [97], a large body of neuroscientific work has robustly identified a principle of functional specialization within the higher-level visual cortex [98,99,100]. This well-documented phenomenon, commonly referred to as category-selectivity, posits that distinct neural populations exhibit strong preferential responses to specific categories of stimuli. It has been instrumental in shaping our understanding of object recognition in the brain. Seminal studies have identified several such category-selective areas, primarily within the ventral visual stream—often dubbed the "what" pathway for its role in object identification. Among the most widely-replicated of these specialized regions are:

- **Body-Selective Regions:** The Extrastriate Body Area (EBA) and Fusiform Body Area (FBA) show preferential activation to images of human bodies and body parts over other object categories [100].

- **Face-Selective Regions:** A network of regions, including the Occipital Face Area (OFA) and the Fusiform Face Area (FFA), responds robustly to faces. The OFA is thought to process individual facial features, while the FFA is more involved in processing the holistic configuration of a face [98].

- **Place-Selective Regions:** The Parahippocampal Place Area (PPA) and the Occipital Place Area (OPA) are selectively activated by visual scenes and landscapes. The PPA is particularly sensitive to the spatial layout of a scene, whereas the OPA may be more attuned to local scene elements and navigational affordances [99].

Understanding this established functional architecture is crucial, as it provides a robust ground truth for validating computational models that aim to simulate the brain's visual processing capabilities.

## I.2. Validating Mind-Omni: Replicating Known Category-Selectivity

To establish the neuroscientific validity of Mind-Omni, we first sought to replicate its ability to capture the well-documented category-selectivity of the visual cortex. We focused on three canonical selective categories: bodies, faces, and places. Specifically, we curated corresponding image sets from the NSD test split (approx. 50 images per category). These images, along with their descriptive captions, were fed into the trained Mind-Omni model to predict fMRI responses. The predicted responses were then averaged across all stimuli within each category and projected onto the cortical surface for visualization.

The results, presented in Fig. 17, demonstrate a striking correspondence with established neuroscientific findings. The colormap indicates the predicted activation level, with red signifying higher activation and blue representing lower levels. As shown, when the model was presented with images of human bodies (e.g., athletes in action, a hand holding a phone), we

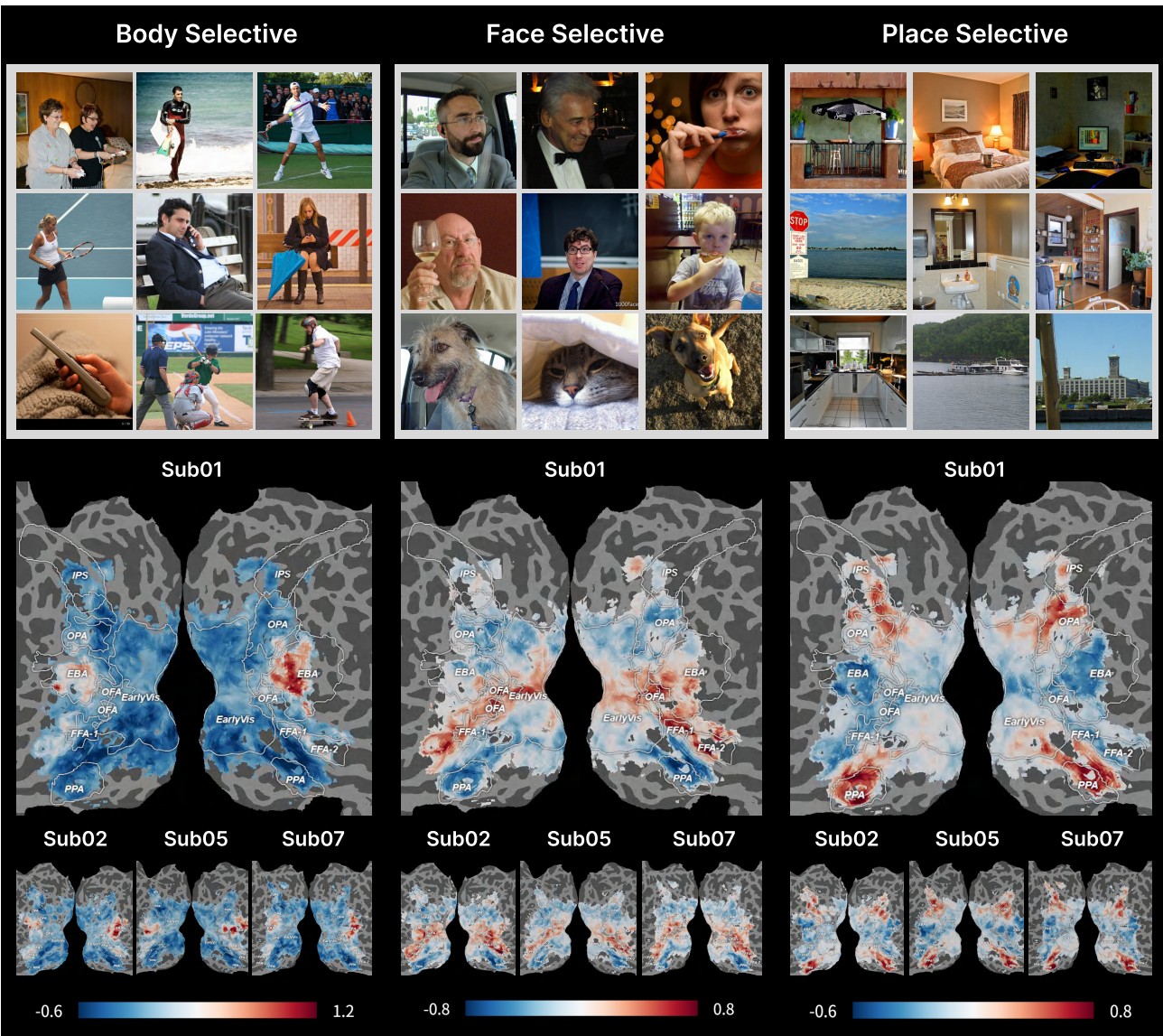

*Figure 17.* Visualization of Mind-Omni's predicted neural responses to category-specific visual stimuli (e.g., Body, Face, Place), examining its learned ROI selectivity. The predicted fMRI responses are projected onto the cortical surface, where red indicates higher activation levels. See Section I.

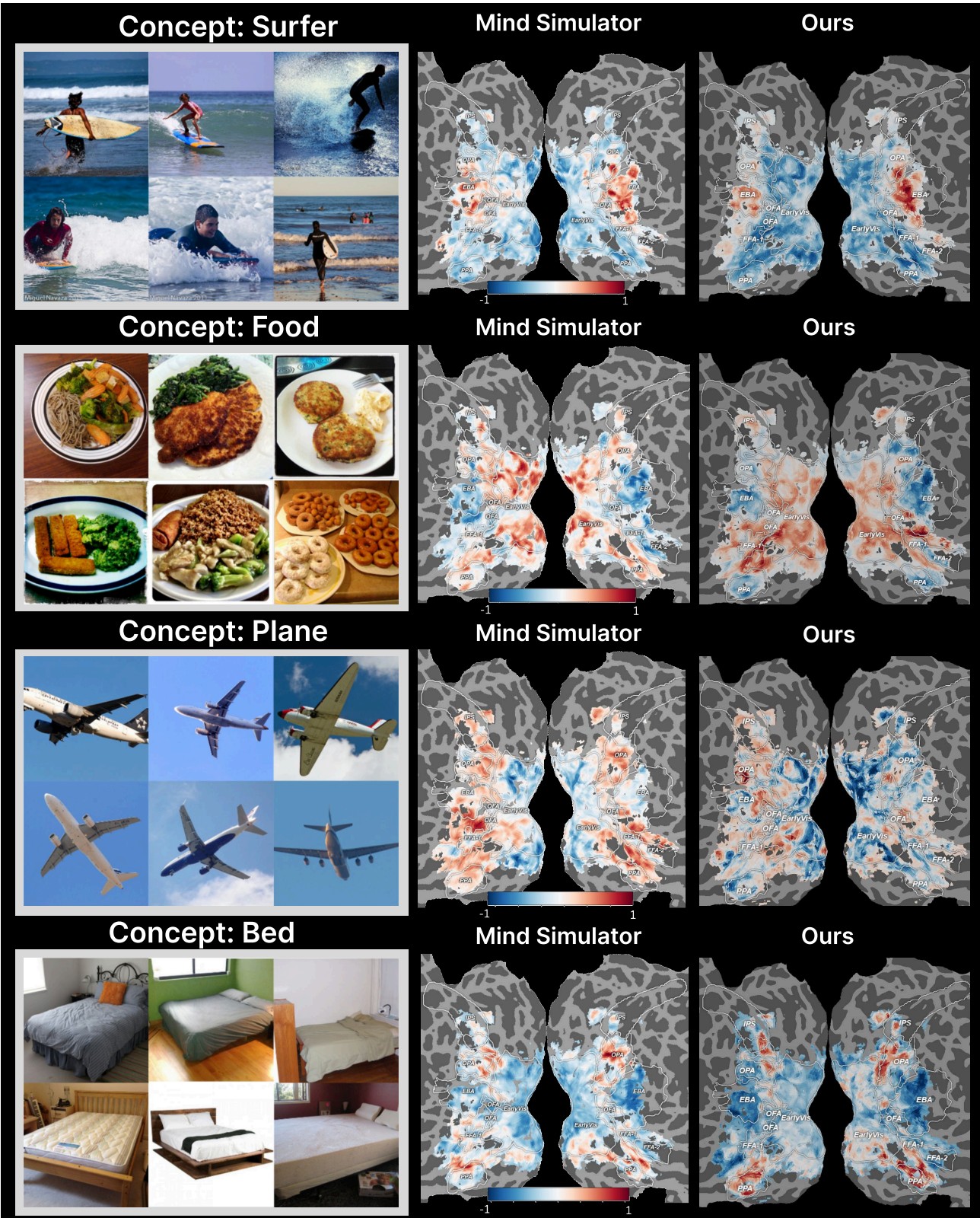

*Figure 18.* Localized concept-selective regions derived from fMRI predictions by Mind-Omni. The predicted neural representations for different concept sets are visualized using the same method as MindSimulator [25]. Results for subject 1 are shown. See Section I.

observed strong and localized activation in the body-selective EBA. Similarly, inputting images of faces—including both human and animal faces—led to pronounced activation in the face-selective OFA and FFA, while other regions remained quiescent. Finally, for scene images such as bedrooms, kitchens, and beaches, the place-selective PPA and OPA were robustly activated.

This consistent pattern of category-selective activation was observed across all four subjects. This outcome confirms that Mind-Omni is not merely performing a superficial numerical fitting. **Instead, it has learned to capture the intricate functional architecture and internal relationships of the visual cortex, including high-level properties like category-selectivity.** This successful replication validates the model's effectiveness and its fidelity as a computational tool for further neuroscientific inquiry.

### I.3. Probing Novel Concept-Selective Regions with Mind-Omni

Having validated Mind-Omni's ability to replicate known neural phenomena, we now employ it to conduct preliminary explorations into the neural representations of more novel concepts.

Following the methodology of MindSimulator [25], we employed CLIP's zero-shot classification capability to identify the top 200 images with the highest semantic specificity for a given concept from the MSCOCO dataset. We then used Mind-Omni to synthesize the corresponding fMRI responses for these image sets. The predicted responses were averaged across samples and projected onto the cortical surface using a consistent colormap, as shown in Fig. 18.

The resulting cortical maps for these novel concepts are remarkably consistent with those generated by the SOTA model, MindSimulator [25], demonstrating our model's powerful capacity to capture concept-level information in fMRI signals. Furthermore, the visualizations reveal an insightful pattern: for high-level concepts such as 'Surfer' or 'Plane', the predicted activations are not confined to a small, isolated patch of cortex. Instead, they are broadly distributed across the higher-level visual cortex. This observation aligns with the principle of distributed processing in the brain, where complex concepts are not encoded by single neurons or voxels but by the coordinated activity of multiple, spatially distant brain regions [76,77,78]. This distributed architecture is thought to be crucial for the brain's efficiency and robustness in processing diverse information.

The exploratory results presented in this section underscore the immense potential of Mind-Omni as a computational testbed for investigating the frontiers of neural information processing.

## J. Limitations and Future Work

While our work introduces the first unified framework for tri-modal brain-vision-language modeling and establishes a new SOTA for multi-task encoding-decoding, a performance gap remains when compared to leading single-task specialist models. This is particularly evident in high-fidelity tasks such as image reconstruction. We attribute this discrepancy to the current scale of our training data and model parameters, which may not yet fully unlock the framework's latent potential.

Our future work will address these limitations along two primary axes. First, we plan to scale up the model by substantially increasing the number of trainable parameters. Second, we aim to curate a larger and more diverse neuroimaging dataset, expanding beyond fMRI to encompass other critical modalities such as EEG, MEG, and ECoG. Through these concerted efforts, our ultimate vision is to construct a true, large-scale foundation model for the field of neural encoding and decoding.

