# OpenReview forum: "Mind-Omni: A Unified Multi-Task Framework for Brain-Vision-Language Modeling via Discrete Diffusion"
_ICML.cc/2026/Conference — ICML 2026 spotlight_

### Official Review · Reviewer_wT4g · 2026-03-06

**Soundness:** 3
**Presentation:** 3
**Significance:** 3
**Originality:** 3
**Overall Recommendation:** 5
**Confidence:** 5

**Summary:**

The authors aim to investigate the problem of building a single model that can both encode and decode across modalities, unifying seven tasks via a discrete diffusion backbone plus a brain tokenizer that discretizes fMRI into semantically aligned tokens, and a brain question-answering instruction-tuning dataset to support reasoning-style outputs.

**Compliance With Llm Reviewing Policy:**

Affirmed.

**Final Justification:**

I have read the authors' replies and additional experiments and believe they've addressed my concerns well. Overall I believe my original score already reflected the good quality of the paper and maintain my position.

**Key Questions For Authors:**

1. What exactly do you mean by “expert models” in this paper, and why does Table 1 not reflect recent multi-task trends (including instruction-tuned or otherwise unified models from the last two years)?
2. Can you provide a clearer and more concrete motivation for diffusion over AR here? In particular, please explain the confounding-bias claim in plain terms and justify why the same conditioning and modality shuffling would not work comparably in an AR backbone.
3. How do you handle causality and directionality across tasks, especially given that image–caption pairs are non-causal, while brain tasks can be asymmetric (stimulus-to-brain vs brain-to-language production)? What role could instruction tuning play in ensuring the desired causal direction at inference time?
4. Why is explicit CLIP-based alignment in the brain tokenizer necessary if the backbone is strong enough? Recent vision–language work suggests that some prior alignment stages can be dropped and learned end-to-end. Do you have evidence that your alignment losses remain beneficial in that regime, or that they are particularly needed for diffusion versus AR?
5. Can you add cross-subject evaluation? Given the small dataset, cross-subject generalization seems essential for judging whether the unified model is learning transferable structure versus subject-specific patterns.

**Limitations:**

The paper discusses potential negative societal impact in terms of mind-reading concerns, but it should more directly discuss methodological limitations: small-data regime relative to the scope of tasks, the need for cross-subject evaluation, and how much the tokenizer’s explicit alignment objectives constrain or shape downstream behavior.

**Strengths And Weaknesses:**

### Soundness

**Strengths**

- The approach is clean, general, and simple.
- Results are convincing and the evaluation is extensive.
- Ablations are good and useful.
- The semantic alignment idea used for the fMRI tokenizer is appealing, and the alignment losses may have broader implications for other brain modalities.

**Weaknesses**

- The diffusion motivation is not well supported in the main text. The paper’s stated motivation around avoiding confounds from a prescribed generation order is not clearly explained, and one sentence in particular is hard to interpret (the claim about confounding bias obscuring true synergistic relationships). This needs a clearer, more concrete argument and, ideally, evidence. The same target-conditioning and modality shuffling setup seems applicable to autoregressive (AR) models as well, which further weakens the case that diffusion is necessary.
- A crucial consideration for unifying encoding and decoding is causality. Paired multimodal datasets (image and caption) are not causal with respect to each other, since either can be generated first. Brain settings add sharper asymmetries: providing language as stimulus versus producing language can share some representations but will also differ in important ways. This matters for claims about unified modeling and for how conditioning is defined. If the goal is stimulus decoding (what stimulus produced the brain activity), this asymmetry is less problematic, but instruction tuning becomes important for making the decoding behavior reliable and well specified.

### Presentation

**Strengths**

- Figures are clear and helpful overall.

**Weaknesses**

- Table 1 and Figure 1 are too small and should be enlarged for readability.
- The expert model framing is not accurate as written. Multi-task models have been widely explored over the last two years (instruction tuning, downstream representation use, fine-tuning, etc.). It is odd that these are not reflected in Table 1 or discussed carefully in the positioning.

### Significance

**Strengths**

- A unified encoder–decoder model spanning brain, vision, and language is a meaningful direction with clear potential utility, especially if it can reduce fragmentation across tasks and enable more systematic study of transfer.

**Weaknesses**

- The dataset appears small for the breadth of claims, which raises concerns about whether observed synergies and generality will hold outside this setting.

### Originality

**Strengths**

- Adapting a unified discrete diffusion backbone with a brain tokenizer and multi-task training over several encode/decode directions is a coherent and novel synthesis.

**Weaknesses**

- Several of the claimed advantages of diffusion over AR (modality permutation invariance, avoiding fixed causal structure) do not feel compelling given that modern AR systems can interleave modalities and preserve ordering when needed. Conceptually, the challenge here is similar to training a text–video model that works both for text-to-video and video-to-text.

---

> ### Author Rebuttal · Authors · 2026-03-29
>
> Thank you for the positive assessment and the very helpful suggestions. Due to the rebuttal length limit, we placed the added tables and figures in the anonymous supplement: **[OSF link](https://osf.io/8rxy2/overview?view_only=f85af5c06ce24aab93febaa1b0cbda0c)**, file **`Reviewer_wT4g.pdf`**. **Below we summarize the main findings.**
>
> **1. “Expert models” / Table 1 positioning.**
> We agree that **“expert models”** is imprecise and will revise it to **task-specific or limited-scope models**. Table 1 already includes some multi-task work (e.g., **BraVL**), but should better reflect recent multi-task / instruction-tuned trends. In the revision, we will expand Table 1 and related work to include more recent unified models (e.g., **One-LLM**,**Brain-Omni**, **NeuroLM**)
>
> **2. Why diffusion instead of AR?**
> We agree that both **AR** and **diffusion** can support the same condition–target setup. Our point is **not** that AR is impossible here, but that they differ in **joint generation (B→I&T)**. In diffusion, image and text tokens interact **synchronously** through full attention at every denoising step. In AR, joint decoding must choose an order (image-first or text-first), so the later modality can attend to the earlier one. This introduces a **directional bias**: an observed gain may partly come from generation order rather than genuine cross-modal cooperation. Diffusion is thus a more natural, order-symmetric testbed for studying synergy. We will rewrite this more clearly and avoid overstating necessity over AR.
>
> **3. Causality, directionality, and instruction tuning.**
> We agree that **image–caption pairs are paired rather than strictly causal**: the same pair can support image→text or text→image, so the data do not define a unique direction. In contrast, **brain tasks are more directional and asymmetric**: stimulus→brain and brain→language may share some representations, but are not interchangeable. Our framework does **not** claim that all tasks share the same causal mechanism; instead, it unifies them under a shared **conditional generation interface**, where directionality is defined by the **condition set / target set** and the corresponding **attention masking**. With modality order **[text, image, brain]**, B→I&T keeps brain clean while adding noise to text/image, whereas I&T→B keeps text/image clean and adds noise to brain. Missing-modality settings (e.g., B→I) are handled by masking the absent modality. **Instruction tuning** is especially important for **BQA**, because it specifies the desired inference behavior: given brain signals and a question, the model should produce a stable **question→answer** output rather than free generation. We will clarify this in the revision.
>
> **4. Why explicit CLIP-based alignment?**
> Even with a strong backbone, the gap between **text**, **image**, and **brain** is large. End-to-end learning may be feasible at very large scale, but our setting is still **data-constrained**. Here, explicit alignment provides a crucial inductive bias. This is supported by **Tab. 5**: removing semantic alignment causes retrieval to drop close to chance and also substantially reduces codebook utilization.
>
> **5. Cross-subject evaluation.**
> We added a **leave-one-subject-out** experiment by holding out **subject 1**. We retrained the tokenizer on the remaining 7 subjects; due to time/compute limits, the main model was retrained for **Stage 1** only. The tokenizer, encoding, and decoding results are reported in **Tables 1–3** of **`Reviewer_wT4g.pdf`**. As expected, performance drops under strict held-out-subject evaluation, but **semantic-level metrics are preserved better than low-level metrics**, and performance remains competitive with or better than **MoPoE / BraVL** on key metrics. Simple linear/MLP baselines can be competitive on some voxel-level measures, while our model is stronger on semantic-level metrics, suggesting better transferable semantic structure.
>
> **6. Small-data regime / scaling.**
> We agree that the current dataset is small relative to the breadth of the claims. To partially address this during rebuttal, we added a simple **data-scaling** study on NSD: we trained the tokenizer with **25% / 50% / 75%** of the training data, and trained the Stage-1 model with **50% / 75%** data. The tokenizer/encoding/decoding curves in **Fig. 1** of **`Reviewer_wT4g.pdf`** show clear improvement with more data, suggesting that the current model is still **data-limited rather than saturated**. Broader cross-dataset scaling remains future work.
>
> **7. Presentation and limitations.**
> We will enlarge **Table 1** and **Figure 1** in the camera-ready version. We will also strengthen the methodological limitations discussion, especially regarding the **small-data regime**, the need for more systematic **cross-subject evaluation**, and how the tokenizer’s explicit alignment objectives may shape downstream behavior.
>
> Thank you again for the constructive feedback.

---

> > ### Author Rebuttal · Reviewer_wT4g · 2026-03-31
> >
> > Thank you for the detailed answers and analysis. My original score reflects the strength of this paper.

---

> > > ### Author Response · Authors · 2026-04-01
> > >
> > > Thank you very much for your careful reading, insightful suggestions, and strong support for our work. We are very glad that our rebuttal addressed your concerns, and we sincerely appreciate your recognition of the paper’s overall strength. Your feedback has been extremely valuable, and we will incorporate the suggested clarifications and revisions in the final version.

---

### Official Review · Reviewer_QkbA · 2026-03-11

**Soundness:** 3
**Presentation:** 3
**Significance:** 2
**Originality:** 2
**Overall Recommendation:** 4
**Confidence:** 4

**Summary:**

This paper proposes a unified brain–vision–language framework (Mind-Omni), that handles both neural encoding and decoding tasks within a single discrete diffusion model.

**Compliance With Llm Reviewing Policy:**

Affirmed.

**Key Questions For Authors:**

- Given the method is well trained, is that possible to test the model's ability on a heldout subject?
- Any error bar/std? It's nice to see some stats analysis.
- Since the model can do encoding, is that possible to do encoding on unseen image concept/category? We could further explore how the pixels/color/concepts are grounded to the fMRI voxels. This is also a good way to validate the model's encoding ability.

**Limitations:**

Need more limitations in the discussion.

**Strengths And Weaknesses:**

Strengths
- Unified framework for encoding and decoding.
- Clear tasks and benchmarking.
- Strong model performance.
- Includes neural analysis showing category-selective maps and concept-selective patterns.

Weaknesses
- (1) The author mentions "We therefore align with the discrete diffusion paradigm for their permutation-invariant nature." Diffusion model is is not controllable for decoding, and further analysis or testing is necessary for this point.
- (2) Is there anyway to heldout one subject out and test model generalization on it. Does the "foundation" brain encoding and decoding model work well than simple baselines, e.g. ridge regression, MLP.

---

> ### Author Rebuttal · Authors · 2026-03-29
>
> Thank you for the positive assessment and constructive suggestions. We have added the new tables and cortical maps to the anonymous supplement: [OSF link](https://osf.io/8rxy2/overview?view_only=f85af5c06ce24aab93febaa1b0cbda0c), file **`Reviewer_QkbA.pdf`**. **Below we summarize the main findings.**
>
> **1. Diffusion Motivation & Controllability**
>
> By “permutation-invariant nature,” we mean **insensitivity to modality order**, not insensitivity to arbitrary perturbations. Since discrete diffusion uses masked conditional prediction, `[img, txt, brain]` and `[brain, img, txt]` define the same task, unlike autoregressive models that usually require a fixed ordering. This is particularly useful for **simultaneous image-text decoding from the same brain input**, which lets us study whether joint decoding yields mutual benefit. More importantly, image and text are updated **synchronously** during denoising, so neither modality is forced to be generated first. This avoids introducing an order-induced directional bias, making diffusion a more natural testbed for examining whether the observed gains come from genuine cross-modal cooperation rather than from one modality having a privileged generation order.
>
> To address stability concerns, we additionally evaluated image decoding and neural encoding on **subject 1 with 5 random seeds**. **Tables 1–2** in `Reviewer_QkbA.pdf` show limited variation, indicating stable performance under sampling randomness.
>
> **2. Held-Out Subject Generalization & Simple Baselines**
>
> We added a **leave-one-subject-out** experiment by holding out **subject 1** and retraining the tokenizer and diffusion backbone on the other 7 subjects. Due to rebuttal-time constraints, we report **Stage-1** results only. The results in **Tables 3–5** show an expected drop on the unseen subject, but our model still outperforms **BraVL** and **MoPoE**. Notably, the drop is much larger on **pixel/voxel-level metrics** than on **semantic-level metrics**, suggesting that the model preserves transferable semantic information across subjects.
>
> We also compared against **Linear** and **MLP** baselines in the held-out-subject setting. As shown in **Table 4**, simple baselines can be stronger on some voxel-level metrics, while our model performs better on **three semantic-level metrics**. We interpret this as evidence that simple baselines fit low-level numerical correlations more easily, whereas our model captures more generalizable semantic structure. For zero-shot decoding, retraining full baselines was too expensive during rebuttal, so we additionally report a lightweight tokenizer-level retrieval comparison; **Table 3** supports stronger cross-subject alignment for our method.
>
> **3. Error Bars / Statistical Analysis**
>
> We added **mean ± std** over the **4 test subjects** in **Tables 6–8** of `Reviewer_QkbA.pdf` to better show cross-subject stability and support more complete future comparisons.
>
> **4. Unseen Concept/Category Encoding**
>
> To probe OOD generalization, we tested unseen stimuli and visualize the predicted cortical maps in **Fig. 1** of `Reviewer_QkbA.pdf`:
> (1) **pixel images** from NSD-OOD [1],
> (2) **grayscale scene images** from NSD-OOD,
> (3) rare concepts in NSD, including **insects** and **tools**, from THINGS [2].
>
> For the first two cases, the results are consistent with neuroscience priors: pixel-art mainly activates **early visual cortex**, suggesting sensitivity to low-level visual structure, while unseen grayscale scenes still activate **OPA/PPA** in addition to early visual cortex, indicating that the model captures scene/layout information beyond color cues.
>
> For **insects** and **tools**, we view the results as **exploratory rather than definitive localization claims**. Insect images induce stronger responses in higher ventral regions such as **EBA/FFA**, which may reflect transfer to animate or biological-form-related representations. Tool images show responses in **IPS** and some higher visual regions, plausibly related to tool/action-affordance structure. Since direct ground-truth cortical maps for these unseen concepts are unavailable, we interpret these results as **preliminary evidence of structured OOD generalization**.
>
> **5. Discussion of Limitations**
>
> We agree that the limitations should be strengthened. We will revise the paper to emphasize that:
> (1) the current study is still conducted in a **data-constrained regime**, and larger-scale data/model scaling as well as extension to other modalities remain important future directions;
> (2) a **unified model** does not yet surpass specialized single-task models on every metric, especially some low-level fidelity metrics;
> (3) the current **held-out-subject** and **OOD concept** analyses are encouraging but still preliminary.
>
> Thank you again for the helpful suggestions.
>
> [1] Gifford et al., “A 7 T fMRI Dataset of Synthetic Images for OOD Modeling of Vision,” Nat. Commun.
>
> [2] Hebart et al., “THINGS-data,” eLife.

---

> > ### Author Rebuttal · Reviewer_QkbA · 2026-04-01
> >
> > I appreciate the authors' effort in providing a thorough rebuttal. I am pleased to note that my prior concerns regarding the statistical analysis and subject generalization have been resolved. However, the current evaluation of the OOD concept relies on qualitative results, whereas quantitative validation is needed. Furthermore, the authors' concession that the omni-model currently lags behind specialized task models limits the overall impact of the work in its current state.
> >
> > Therefore, I will keep my score.

---

> > > ### Author Response · Authors · 2026-04-01
> > >
> > > Thank you again for the clarification. We are glad that the added **statistical analysis** and **held-out-subject evaluation** addressed your main concerns.
> > >
> > > We also agree that quantitative validation would further strengthen the OOD analysis. At present, this part should still be viewed as a qualitative, exploratory analysis. That said, the current visualizations already show structured and neuroscience-consistent generalization patterns under OOD stimuli, rather than arbitrary activations. A stricter quantitative evaluation is currently difficult because external datasets such as NSD-OOD and THINGS have not yet been aligned to the same cortical template / voxel correspondence as NSD, so there is not yet a standardized and reproducible protocol for direct quantitative comparison. For this reason, we prefer to interpret the current OOD results as exploratory evidence rather than make a stronger quantitative claim. In future work, we plan to pursue template alignment and standardized integration of external fMRI datasets, so that OOD generalization can be evaluated in a more controlled setting.
> > >
> > > Regarding the point that the current omni-model still trails specialized task models, we view this as the current trade-off between task breadth and task-specific specialization, rather than a limitation of the unified direction itself. Under the same data regime, a single model that jointly supports multiple heterogeneous encoding/decoding tasks can reasonably be expected to lag behind dedicated single-task systems on some fine-grained metrics, while still offering clear advantages in versatility, transfer, and unified analysis. Consistent with this view, our added NSD data-scaling study suggests that the current model is still data-limited rather than saturated, indicating meaningful room for improvement as the training scale increases.
> > >
> > > Thank you again for your careful reading and helpful feedback.

---

### Official Review · Reviewer_agQy · 2026-03-12

**Soundness:** 3
**Presentation:** 2
**Significance:** 2
**Originality:** 3
**Overall Recommendation:** 4
**Confidence:** 4

**Summary:**

Mind-Omni is a unified multi-task framework specifically designed for brain-vision-language  modeling. While existing state-of-the-art approaches typically rely on specialized expert models, which are inherently restricted to single tasks, such as decoding fMRI signals into images or encoding visual stimuli into neural representations, this paper introduces a versatile architecture built upon discrete diffusion. By framing seven distinct neural encoding and decoding tasks as conditional masked token prediction, the proposed method integrates these traditionally isolated objectives into a single, cohesive framework. Experiments conducted on the Natural Scenes Dataset demonstrate the effectiveness of this unified approach, validate its robust cross-modal alignment capabilities, and provide compelling evidence that the model successfully captures functional neural architectures, offering a powerful computational testbed for neuroscientific inquiry.

**Compliance With Llm Reviewing Policy:**

Affirmed.

**Final Justification:**

The rebuttal has addressed my main concerns. I appreciate the clarifications provided by the authors. Thus I keep my positive rating.

**Key Questions For Authors:**

1. Codebook Size: The experiments show a decrease in codebook utilization as the size increases. Does this suggest an inherent bottleneck in the information density of fMRI signals?
2. Mechanism of Synergy: In the B→I&T (Brain-to-Image & Text) task, does the generated text directly contribute to the quality of image generation, or is the improvement solely a result of sharing a common latent space?
3. Masking Ratio: You chose a 30% masking ratio for fine-grained alignment. Why did you not experiment with a higher ratio (e.g., 75%), similar to that used in MAE?

**Limitations:**

Yes

**Strengths And Weaknesses:**

**Strengths**

- The proposed framework achieves unification by reformulating all tasks as conditional masked token prediction problems, thereby bypassing the limitations inherent in task-specific generation.
- The authors introduce a Brain Tokenizer that resolves the alignment challenge between continuous brain signals and discrete multimodal data through a multi-level alignment strategy.
- The authors extend their analysis beyond traditional encoding/decoding metrics such as PCC and MSE by exploring qualitative neuroanatomical consistency (e.g., category-selective regions in the visual cortex), which significantly strengthens the persuasiveness of the work.

**Weaknesses**

- Although the authors introduce the Brain Question Answering task, relying solely on traditional text generation metrics like BLEU and ROUGE for inference evaluation is insufficient. In reasoning tasks, subtle semantic deviations can lead to total logical failure, yet such models may still yield high scores under these metrics. These metrics may not accurately reflect the model's true reasoning performance. It is highly recommended that the authors adopt an evaluation framework specifically designed for logical consistency
- In Figures 6 and 18, the qualitative comparison with Mind Simulator fails to strictly follow the principle of controlled variables, as inconsistent background masks are used across methods. This introduces confounding factors that prevent an objective assessment of the model’s generation capability within the core target regions. I strongly urge the authors to use identical background masks for all baselines and their proposed method in the revised manuscript to ensure a fair and rigorous comparison.
- The framework appears to rely heavily on the alignment between brain signals and visual/textual modalities. By projecting brain activity into a latent space already dominated by image/text features, the reasoning process seems to occur primarily within the visual modality rather than the brain signals themselves. The claim regarding brain reasoning might need more careful clarification, as the model essentially delegates the reasoning task to pre-aligned visual representations, potentially bypassing the unique neural dynamics that this study should be capturing.

---

> ### Author Rebuttal · Authors · 2026-03-29
>
> Thank you for your careful reading, support, and insightful suggestions. We have placed the added figures in the anonymous supplement: **[OSF link](https://osf.io/8rxy2/overview?view_only=f85af5c06ce24aab93febaa1b0cbda0c)**, file **`Reviewer_agQy.pdf`**. **Below we summarize the main findings.**
>
> **1. BQA evaluation beyond BLEU/ROUGE.**
> We agree that lexical metrics alone are insufficient for reasoning-style evaluation. We therefore added an **LLM-as-Judge** evaluation using **Qwen3-VL-30B-A3B-FP8**: given the stimulus image, question, reference answer, and model output, the judge determines whether the answer is correct. We use the same prompt and evaluation protocol for all compared methods, asking the judge to assess factual correctness with respect to the image/question/reference. The accuracy values below are **mean±std over 4 test subjects**:
>
> | Methods | LLM-as-Judge |
> | ------- | :----------: |
> | OneLLM  |  19.12±1.23  |
> | UMBRAE  |  25.48±1.17  |
> | Ours    |  24.37±1.14  |
>
> These results show that our method remains competitive under a factuality-aware evaluation protocol and substantially outperforms OneLLM. We will revise the paper to avoid relying on BLEU/ROUGE when discussing BQA.
>
> **2. Controlled visualization for Fig. 6 / 18.**
> We agree that the original qualitative comparison was not sufficiently controlled. To address this, we **reproduced MindSimulator on our data** and re-rendered the compared maps under the **same visualization protocol** with the **same background mask and display setting**. The updated comparisons are provided in **Figs. 1–2** of **`Reviewer_agQy.pdf`**.
>
> **3. Clarifying the “brain reasoning” claim.**
> We agree that the phrase “brain reasoning” was too strong. A more accurate interpretation is that the **pretrained multimodal backbone provides the reasoning prior**, while **brain signals provide indispensable grounding information**. To test this, we added a **shuffled-brain** ablation and again evaluated with **Qwen3-VL-30B-A3B-FP8** (**mean±std over 4 test subjects**):
>
> | Methods        | LLM-as-Judge |
> | -------------- | :----------: |
> | Shuffled Brain |  8.45±1.43   |
> | Ours           |  24.37±1.14  |
>
> Replacing the true brain input with shuffled signals causes a large drop, showing that the answer content is strongly brain-grounded rather than produced from language priors alone. We also provide qualitative examples in **Fig. 3** of **`Reviewer_agQy.pdf`**, showing that without informative brain input, the model can still produce fluent answers by relying on its learned priors and defaulting to the most probable response pattern, but these answers no longer faithfully reflect what the subject actually saw. We will therefore **soften the wording** in the revision.
>
> **4. Codebook size and information bottleneck.**
> Yes—our results suggest that, in the current NSD/fMRI setting, there is a practical information bottleneck. As the codebook size increases, utilization drops substantially, implying that the available brain signals do not support a much larger set of reliably separable discrete states under the present data and training conditions. A plausible explanation is the combination of limited sample size, fMRI noise, and the relatively low effective dimensionality of the signals. In other words, beyond a moderate codebook size, the representation becomes increasingly under-utilized rather than more informative.
>
> **5. Mechanism of Synergy**
> Our current results are **consistent with the benefit of sharing a common latent space**: under the **same brain input**, joint decoding improves both image generation quality and text decoding accuracy, as also evidenced by the quantitative results in **Tables 2 and 3** and the qualitative examples in **Fig. 8** of the main paper. We therefore interpret these gains as arising from an aligned multimodal latent space that supports mutually beneficial interactions between image and text decoding.
>
> **6. Why 30% masking ratio instead of 75% like MAE?**
> MAE uses a high masking ratio because image patches are highly spatially redundant, whereas language/CLIP text token states are much less redundant. We additionally ran a mask-ratio ablation, shown below:
>
> | mask ratio | rPCC ↑ | B2I ↑ | B2T ↑ | codebook usage ↑ |
> | ---------- | -----: | ----: | ----: | ---------------: |
> | 0.15       | 0.66 | 0.64 | 0.53 | 82% |
> | *0.30*     | *0.68* | *0.62* | *0.59* | *80%* |
> | 0.45       | 0.68 | 0.63 | 0.57 | 78% |
> | 0.60       | 0.55 | 0.53 | 0.48 | 40% |
> | 0.75       | 0.52 | 0.42 | 0.40 | 40% |
>
> Performance is similar for **moderate ratios (15%–45%)**, but degrades substantially at **60%–75%**. Thus, **30% is not universally best on every metric, but provides the best overall trade-off** between reconstruction, retrieval, and codebook usage.
>
> Thank you again for the constructive feedback. We believe these new analyses substantially strengthen the paper.

---

> > ### Author Rebuttal · Reviewer_agQy · 2026-04-01
> >
> > I thank the authors for their response. My concerns are fully addressed. Thus I decide to keep my positive rating.

---

> > > ### Author Response · Authors · 2026-04-02
> > >
> > > Thank you very much for your careful evaluation and encouraging feedback. We are pleased that our rebuttal has adequately addressed your concerns, and we sincerely appreciate your positive assessment of the paper. Your comments have been very valuable, and we will incorporate the relevant clarifications and improvements in the final version.

---

### Official Review · Reviewer_YH66 · 2026-03-13

**Soundness:** 3
**Presentation:** 4
**Significance:** 3
**Originality:** 3
**Overall Recommendation:** 5
**Confidence:** 3

**Summary:**

The authors aim to investigate the problem of modeling interactions between external stimuli and internal neural representations. Furthermore, the authors strive to assess a broad aspect of multi-task synergies by proposing Mind-Omni, a unified discrete diffusion framework. Utilizing a novel Brain Tokenizer, it successfully unifies seven encoding and decoding tasks, demonstrating strong SOTA performance.

**Compliance With Llm Reviewing Policy:**

Affirmed.

**Final Justification:**

My concerns are well addressed, and I remain my acceptance score.

**Key Questions For Authors:**

see Weaknesses

**Limitations:**

yes

**Strengths And Weaknesses:**

**Strengths:**
1. Mind-Omni presents the first unified Brain-Vision-Language modeling paradigm capable of supporting seven distinct tasks. By leveraging a discrete diffusion model, it successfully integrates encoding and decoding tasks within a single cohesive framework.
2. To enable the processing of continuous brain signals alongside discrete text and image tokens within a shared representational space, the authors design a Brain Tokenizer equipped with specialized alignment constraints.
3. Given that existing fMRI datasets contain only extremely brief COCO captions and thus lack a detailed semantic grounding, the authors leverage advanced MLLMs to carefully construct a brain-computer instruction-tuning dataset (BQA Dataset).

**Weaknesses:**
1. As illustrated in Figure 4, Mind-Omni underperforms the corresponding task-specific state-of-the-art models on several core tasks, such as B→I and B→T. The authors attribute this performance gap to the limited scale of training data and model parameters. However, no supporting experiments or analyses are provided to substantiate the claim that the proposed framework is scalable.
2. The authors' claim regarding the discovery of synergistic effects arising from multimodal joint training is not convincing. Specifically, when predicting B, simultaneously providing both I and T as inputs inherently supplies the model with a greater amount of information. An improvement in prediction accuracy given increased input information is an expected outcome and does not, in itself, constitute evidence of genuine synergy within the model. This claim lacks support from rigorously designed controlled experiments that could disentangle the effect of additional input information from true cross-modal synergistic gains.

---

> ### Author Rebuttal · Authors · 2026-03-29
>
> Thank you very much for the positive assessment and for highlighting two important issues: **(1) scalability evidence** and **(2) the interpretation of the claimed synergy effects**. We have placed the added figures in the anonymous supplement: **[OSF link](https://osf.io/8rxy2/overview?view_only=f85af5c06ce24aab93febaa1b0cbda0c)**, file: **`Reviewer_YH66.pdf`**. **Below we summarize the main findings.**
>
> **(1) On scalability.**
>
>  We agree that the original submission did not provide direct scaling evidence, and thank you for pointing this out. Due to the rebuttal timeline, we were unable to run full scaling over larger datasets or larger backbones. As a practical but informative proxy, we added a **data-scaling study on NSD** by training the **Brain Tokenizer** with **25% / 50% / 75% / 100%** of the training data, and then evaluating both tokenizer quality and a **Stage-1 model** trained on top of it. The curves are provided in the anonymous supplement: **[OSF link](https://osf.io/8rxy2/overview?view_only=f85af5c06ce24aab93febaa1b0cbda0c)**, file: **`Reviewer_YH66.pdf`**. Across all tested fractions, **both tokenizer performance and Stage-1 downstream performance improve monotonically with more data, and we do not observe clear saturation at 100%**. While this does not yet constitute a full scaling-law study, it provides direct evidence that the current framework remains **data-limited rather than saturated**. This is also consistent with our existing ablations: in the paper, stronger training recipes and better caption data already yield consistent gains, suggesting that the framework benefits from improved data and optimization rather than being intrinsically bottlenecked. We will clarify this scope in the revision and explicitly state that broader data/model scaling remains an important direction for future work.
>
> **(2) On the “synergy” claim.**
>
>  We agree with your concern that, for **encoding** (I&T→B), improved performance over I→B or T→B alone can be partly attributed to having more input information. We will therefore **revise our wording** and describe the encoding result more carefully as **cross-modal complementarity**, rather than claiming it alone as definitive evidence of synergy. In particular, Fig. 7 shows that image-only and text-only conditions emphasize different cortical regions, while the joint condition yields broader and stronger prediction across visual cortex, which we interpret as complementary integration of visual and semantic information.
>
> We believe the **stronger evidence for synergy comes from decoding**, where the input is held fixed as the **same brain signal B**. Under this matched-input setting, **joint decoding (B→I&T) outperforms the corresponding single-task variants on both modalities simultaneously**: for example, in Table 2 the high-level image metric **CLIP improves from 66.7 (B→I) to 79.8 (B→I&T)**, and in Table 3 the text metric **ROUGE improves from 13.35 (B→T) to 30.54 (B→I&T)**. Qualitatively, Fig. 8 further shows that joint decoding produces captions with richer visual details while reducing artifacts. Since the conditioning input is identical, this improvement cannot be explained by “more input information”; instead, it suggests beneficial interaction between the two decoding objectives within the unified model.
>
> We appreciate this helpful suggestion and will revise the manuscript accordingly by:
>  (i) **softening the encoding-side claim to complementarity**, and
>  (ii) **emphasizing the matched-input decoding result as the main evidence for inter-task synergy**.
>
> Thank you again for the constructive feedback.

---

> > ### Author Rebuttal · Reviewer_YH66 · 2026-04-03
> >
> > I think the response has solved my concerns. So, I remain my acceptance score.

---

> > > ### Author Response · Authors · 2026-04-03
> > >
> > > We sincerely thank the reviewer for the careful reconsideration of our rebuttal and for confirming that the concerns have been fully resolved; we greatly appreciate your constructive feedback and your support of our work.

---

### Decision · Program_Chairs · 2026-04-30

**Decision:**

Accept (spotlight)

**Comment:**

This interdisciplinary work provides a diffusion-based framework with the goal to unify a number of distinct encoding and decoding tasks for transforming signals from neural data into discrete tokens, supporting the following modalities: brain signals, images, and text. This is a strong submission which is likely to interest readers from machine learning, neuroscience and brain-computer interfaces. Reviewers unanimously recommend acceptance and indicated that their concerns are fully resolved.